# How iteration composition influences convergence and stability in deep learning

**Benoit Dherin**[*]                                    *dherin@google.com*
*Google Research*

**Benny Avelin**[*]                                     *benny.avelin@math.uu.se*
*Department of Mathematics, Uppsala University*

**Anders Karlsson**[*]                                  *anders.Karlsson@unige.ch*
*Department of Mathematics, University of Geneva and Uppsala University*

**Hanna Mazzawi**                                       *mazzawi@google.com*
*Google Research*

**Javier Gonzalvo**                                     *xavigonzalvo@google.com*
*Google Research*

**Michael Munn**                                        *munn@google.com*
*Google Research*

**Reviewed on OpenReview:** *https://openreview.net/forum?id=GZCBM2Yo3a*

## Abstract

Despite exceptional achievements, training neural networks remains computationally expensive and is often plagued by instabilities that can degrade convergence. While learning rate schedules can help mitigate these issues, finding optimal schedules is time-consuming and resource-intensive. This work explores theoretical issues concerning training stability in the constant-learning-rate (i.e., without schedule) and small-batch-size regime. Surprisingly, we show that the composition order of gradient updates affects stability and convergence in gradient-based optimizers. We illustrate this new line of thinking using backward-SGD, which produces parameter iterates at each step by reverting the usual forward composition order of batch gradients. Our theoretical analysis shows that in contractive regions (e.g., around minima) backward-SGD converges to a point while the standard forward-SGD generally only converges to a distribution. This leads to improved stability and convergence which we demonstrate experimentally. While full backward-SGD is computationally intensive in practice, it highlights that the extra freedom of modifying the usual iteration composition by reusing creatively previous batches at each optimization step may have important beneficial effects in improving training. Our experiments provide a proof of concept supporting this phenomenon. To our knowledge, this represents a new and unexplored avenue in deep learning optimization.

## 1 Introduction

In recent years, neural networks have achieved remarkable success across diverse domains from text generation Gemini (2024; 2023); Brown et al. (2020) and image creation Ramesh et al. (2022; 2021); Saharia et al. (2022) to applications in protein folding Jumper et al. (2021a;b) and material discovery Merchant et al. (2023). However, their training remains challenging and compu-

---

[*]These authors contributed equally to this work

tationally expensive. One of the reasons for this is due to training instabilities which often occur Morchdi et al. (2023); Li et al. (2019); Cohen et al. (2021); Chen et al. (2018) and which produces hard to interpret loss curves, wasted computation time, and potentially failed experiments. One way to view this challenge is as a trade-off between stability and performance: hyperparameter settings that often yield better test performance, such as higher learning rates and smaller batch sizes, tend to exacerbate these instabilities.

As a basic example, consider the training batch size. For many common optimizers, smaller batch sizes often lead to improved test performance. In fact, recent research has shown that small batches induce a form of implicit regularization Novack et al. (2023); Smith et al. (2021); Dherin et al. (2022); Keskar et al. (2017); Ali et al. (2020) which benefits generalization. On the other hand, the greater variability of small batches tends to exacerbate oscillations of the training loss, prolonging time to convergence. In this work, we demonstrate that these instability and convergence issues associated with small batch sizes can be mitigated without loss of generalization power by reversing the composition order used to produce each iterate. The general line of thinking behind our approach consists of leveraging not only the current batch, but also the previous batches to produce the current parameter iterate. The creative reuse of previous batches at a given step has been already explored in Choi et al. (2019) for instance in order to speed up training by composing previous batch gradient updates to the current iterate while waiting for the data pipeline to deliver a new batch in case of IO bound situations. In this work, we reverse the composition order of all the batches received so far from initialization and show theoretically and practically that the sequence of iterates produced this way enjoys better convergence and stability properties than the usual composition order of the batch gradient updates. We now describe below this reverse composition procedure in details.

Namely, standard training algorithms, such as stochastic gradient descent (SGD), Adam, and other gradient-based optimizers, are iterative processes. At each step, these algorithms update the network parameters $\theta$ using a randomly sampled data batch $B_i$. This update can be formalized as a transformation, $\theta' = T_i(\theta)$, where $\theta'$ represents the new parameter value. Because at each step $i$ the batch $B_i$ is randomly sampled, the update operator $T_i$ can be formalized as a random operator. For instance, in the case of SGD, the random operator is defined by the update rule $T_i(\theta) = \theta - h\nabla L_{B_i}(\theta)$, where $L_{B_i}$ is the loss function evaluated on batch $B_i$ and $h$ is the learning rate. The sequence of parameter updates generated by these iterations, starting from an initial parameter value $\theta$, defines the standard learning trajectory, which we will refer to as the *forward trajectory*:

$$\theta_0 = \theta, \quad \theta_1 = T_1(\theta), \quad \theta_2 = T_2 T_1(\theta), \quad \ldots, \quad \theta_n = T_n T_{n-1} \cdots T_1(\theta).$$

**Notation:** To conserve notation, we write $T_i T_j$ to denote the mapping composition $T_i \circ T_j$; i.e., $T_i T_j(\theta) = T_i(T_j(\theta))$.

Moderate or large learning rates and small batches, when used with the standard forward trajectory for training, tend to destabilize common gradient-based optimizers, causing convergence and stability issues. Consequently, learning rate schedules have become essential as an attempt to encourage proper convergence of the loss during training. Our key contribution, which we support with both theoretical analysis and empirical results (see Figure 2 and Appendix A), is in demonstrating that reversing the batch composition order to produce an iterate at a given step–what we call the *backward trajectory*–leads to significantly more stable convergence. Specifically, the backward trajectory consists of iterates generated by composing the training batches in reverse order; i.e.,

$$\theta_0 = \theta, \quad \theta_1 = T_1(\theta), \quad \theta_2 = T_1 T_2(\theta), \quad \ldots, \quad \theta_n = T_1 T_2 \cdots T_n(\theta).$$

For the sake of clarity, let us work out the first two iterates both for forward and backward. For instance for SGD, if we receive first the batch $B_1$ and then the batch $B_2$ in sequence in the training loop, then at the first step both the backward and forward iterates coincides with $T_1(\theta) = \theta - h\nabla L_{B_1}(\theta)$, where $\theta$ is the randomly initialized parameter value. However at the second step the forward iterate becomes

$$\begin{aligned} T_2 T_1(\theta) &= T_1(\theta) - h\nabla L_{B_2}(T_1(\theta)) \\ &= \theta - h\nabla L_{B_1}(\theta) - h\nabla L_{B_2}(\theta - h\nabla L_{B_1}(\theta)). \end{aligned}$$

Figure 1: Naive implementation of the backward dynamics: Forward iterations (left) and backward iterations (right). The training steps are represented by Pac-men consuming batches. Forward iterations maintain a training state and consume a new batch at each step, while backward iterations restart the training and consume all the batches received so far in reverse order.

On the other hand, the second step of the backward iterate is

$$\begin{aligned} T_1 T_2(\theta) &= T_2(\theta) - h\nabla L_{B_1}(T_2(\theta)) \\ &= \theta - h\nabla L_{B_2}(\theta) - h\nabla L_{B_1}(\theta - h\nabla L_{B_2}(\theta)). \end{aligned}$$

A naive (and computationally intensive) implementation of the backward optimization is depicted in Figure 1 where all the batches received so far at each step are re-processed from scratch in the reverse order in which they were received from the initialization point.

**Main contributions:** The main contributions of this paper are to show theoretically (Theorems 2.2 and 2.6) and experimentally (Figure 2 and Appendix A) that backward optimization has better convergence and stability properties than the standard forward optimization. As already known (see related work in Section 1.1), forward trajectories do not generally converge toward points but rather toward a probability distribution of the iterates (in the fixed learning-rate regime).

We argue that the advantage of backward trajectories comes from their convergence toward actual points (see Theorem 2.2) sampled from the forward distribution (see Theorem 2.6). We prove this using a generalization of the Banach fixed point theorem when the random maps $T_i$ become contractions. Note that backward optimization can theoretically be applied to any gradient-based optimizer (not only to SGD) to improve stability and convergence, since our theoretical statements hold at the level of the $T_i$'s, no matter what their actual forms are. To our knowledge, these concepts are new in deep learning optimization. The main goal of this paper is to expose how this phenomenon manifests in deep learning. We defer engineering applications leveraging this phenomenon (like efficient implementations of the backward SGD) to future work, while outlining a few potential directions at the paper's conclusion.

Overall, this work should be understood as a proof of concept showing that reversing the iteration composition order can significantly improve stability and convergence in stochastic optimization.

## 1.1 Related work

**Convergence of SGD:** A number of works prove under different assumptions that the (forward) iterates of SGD with constant learning rate do not converge toward points but rather toward a stationary probability distribution: Merad & Gaïffas (2023) and Dieuleveut et al. (2020) show this for the strongly convex case; Shirokoff & Zaleski (2024) treats the non-convex case with separable loss; Babichev & Bach (2018) focus on

losses coming from exponential family models; Cheng et al. (2020) quantify the rate of convergence of SGD to its stationary distribution in non-convex optimization problems; see also Dieuleveut & Bach (2016) and Meyn & Tweedie (1993). In Huang et al. (2017), the authors leverage the fact that SGD with fixed large learning rate oscillates between different solutions in order to create a cheap average of models by saving the explored parameters along the way. For convergence to a particular solution, the forward order for SGD needs an extra decaying learning rate schedule as shown in Robbins & Monro (1951) or Mertikopoulos et al. (2020).

**Contractions in deep learning:** From our point of view, one important feature that leads to convergence for the backward trajectory is the contraction property of the random operators. This notion (which we believe is under-exploited in deep learning) has surfaced in different contexts in deep learning: see Bergomi et al. (2019), Qian & Wegman (2019), and Avelin & Karlsson (2022).

**Markov chains, iterated functions, and MCMC:** In many Markov Chain Monte Carlo (MCMC) algorithms the goal is to sample from a distribution $\mu$. The idea is to construct a Markov chain with stationary distribution $\mu$ and then run the chain for a long time to get samples from $\mu$. The Propp-Wilson algorithm uses a form of backward iterations to accelerate the convergence toward samples from the distribution (see Propp & Wilson (1996)). More generally, the idea of backward dynamics is hidden in many constructions in Markov chain theory when the Markov chain is given by iterations of random operators as illustrated in Diaconis & Freedman (1999). In particular, they prove a general result concerning the convergence in distribution of these types of iterated Markov chains using the backward dynamics.

**Stability in deep learning:** Already instability issues appear in the full-batch regime, and a number of theoretical works have studied it under the heading *edge of stability* Cheng et al. (2020); Wu et al. (2024); Cai et al. (2024). Other works have also studied stability in the large batch regime after a batch size saturation takes place using the implicit conditioning ratio Lee et al. (2022); Agarwala & Pennington (2024). In this work we focus on the stochastic or small batch setting. Our findings do not really matter for the full-batch setting since backward and forward iterates then coincide. In the context of physics-informed neural networks, it has been observed that the gradient field is stiff, producing instabilities in learning trajectories Wang et al. (2020). To remedy this Li et al. (2023) propose a backward Euler scheme to stabilize training in this context. However the backward Euler method is an implicit Runge-Kutta method, which is different from using iteration backward.

**Sample order:** Curriculum learning Soviany et al. (2022) leverages the impact of sample order for generalization Mange (2019) by organizing training examples in a meaningful sequence, typically starting with simpler examples and gradually introducing harder ones, thereby optimizing the learning process and improving model performance. The backward optimization can be viewed as an automated form of curriculum, mitigating the forgetting of previous examples as new examples are added; a phenomenon which is also related to catastrophic forgetting and the stability gap in continual learning (see Lange et al. (2023) for instance).

## 2 A backward contraction principle

The contraction mapping principle, also called the Banach fixed point theorem, is a cornerstone result in mathematics and science, in particular for finding solutions of equations (Newton's method) or of differential equations (Picard's method of successive approximation). It is also behind Google's PageRank algorithm Page et al. (1998). It concerns the existence of a fixed point of maps which are uniform contractions of a complete metric space[1]. In detail, let $(\Omega, d)$ be a complete metric space and $T : \Omega \to \Omega$ be a continuous map. If there exists $0 \leq k < 1$ such that

$$d(T(\theta_1), T(\theta_2)) \leq k \cdot d(\theta_1, \theta_2),$$

---

[1]Recall that a complete metric space $(\Omega, d)$ is a set $\Omega$, equipped with a distance metric $d(\theta_1, \theta_2)$ for which all Cauchy sequences (i.e., sequences $\theta_n$ such that $d(\theta_n, \theta_m) \to 0$ as $n, m \to 0$) converge to points in the space; typically $\mathbb{R}^d$ with the Euclidean distance is a complete metric space.

for all $\theta_1, \theta_2 \in \Omega$, then $T$ is called a *uniform contraction*. The fixed point theorem now states that $T$ has a fixed point, that is, $\theta_0 \in \Omega$ such that $T(\theta_0) = \theta_0$. This point can be found by iterating the map:

$$T^n(\theta) \to \theta_0$$

for any $\theta \in \Omega$ as $n \to \infty$. For example, in PageRank Page et al. (1998), one iterates the PageRank matrix from 50 to 100 times and the result is a very good approximation of the PageRank vector $\theta_0$. Another application is the standard proof of convergence for (full-batch) gradient descent around a minimum, which can be interpreted as an application of the Banach fixed point theorem as detailed in the next example:

*Example* 2.1. **(Full-batch gradient descent convergence.)** In this case the operator is $T(\theta) = \theta - h\nabla L(\theta)$ for a loss function $L$. The idea of the proof is to choose the learning rate $h$ small enough so that $T$ becomes a contraction. More precisely, around a minimum $\theta^*$ the operator $T$ can be approximated using a first-order expansion of the gradient around the minimum as $T(\theta) = \theta^* + (I - hH)(\theta - \theta^*)$, where $H = \nabla^2 L(\theta^*)$. Now we have that

$$\|T(\theta_1) - T(\theta_2)\| \leq \|I - hH\|_{\mathrm{op}} \|\theta_1 - \theta_2\|, \tag{1}$$

where $\|I - hH\|_{\mathrm{op}}$ is the operator norm of $1 - hH$, that is the operator maximum eigenvalue: $\max_i |1 - h\lambda_i|$ (here the $\lambda_i$'s are eigenvalues of $H$). Now it is easy to verify that $\|I - hH\|_{\mathrm{op}} < 1$ if and only if the learning rate is strictly smaller than $2/\lambda_{\max}$ where $\lambda_{\max}$ is the largest eigenvalue of $H$. Convergence for that setting follows from the Banach fixed point theorem.

The generalization of this convergence argument to Stochastic Gradient Descent (SGD) is actually problematic. The main reason is that now we do not have a single operator but a sequence of them $T_1, T_2, \ldots$, each computing the loss gradient on a different batch of data. Even if each of the operators is a contraction it turns out that usual forward iterations will not converge to a point in general as illustrated in Example 2.4. Theorem 2.2, however, establishes the convergence of the backward iterations.

**Theorem 2.2.** *(Backward contraction mappings principle) Let $T_i$ be a sequence of continuous self-maps of a complete metric space. Assume $T_i$'s are uniform contractions, with a certain $k < 1$ in common, and for some $\theta$ there is a constant $D$ such that,*

$$d(\theta, T_i(\theta)) < D \quad \text{for all } i. \tag{2}$$

*Then for any $\theta \in \Omega$ the backward iterates*

$$\theta_n = T_1 T_2 \cdots T_n(\theta)$$

*converge to a point $\theta^*$ as $n \to \infty$. Moreover, the convergence rate is exponential: i.e, there is a constant $C$ depending on $\theta$ such that*

$$d(\theta^*, T_1 T_2 \cdots T_n(\theta)) \leq C \cdot k^n.$$

*Proof.* Condition (2) expresses that the distance $d(\theta, T_i(\theta))$ is uniformly bounded by a constant $D$ for all $T_i's$ for a point $\theta$. Let us show that if this happens for a single point $\theta$, this happens for all points, provided we change the constant $D$. To see this, take another point $\tilde{\theta}$. We will compute another constant $\tilde{D}$ such that $d(\tilde{\theta}, T_i(\tilde{\theta})) < \tilde{D}$. Namely, using the triangle inequality and the fact that $T_i$ are uniform contractions, we obtain that

$$
\begin{aligned}
d(\tilde{\theta}, T_i(\tilde{\theta})) &\leq d(\tilde{\theta}, \theta) + d(\theta, T_i(\theta)) + d(T_i(\theta), T_i(\tilde{\theta})) \\
&\leq D + (1 + k)d(\theta, \tilde{\theta}) = \tilde{D}.
\end{aligned}
$$

Now, we want to prove that $\theta_n$ is a Cauchy sequence, i.e, that $d(\theta_n, \theta_m)$ tends to zero for $m > n$ as $n \to \infty$. Since $\Omega$ is assumed to be complete, this will mean that the backward iterates $\theta_n$ converge toward a point $\theta^*$. The idea is to bound the quantity

$$
\begin{aligned}
A &= d(\theta_n, \theta_m) \\
&= d(T_1 T_2 \cdots T_n(\theta), \, T_1 T_2 \cdots T_m(\theta)).
\end{aligned}
$$

Because of the backward order (note: the forward order would not allow that), we can apply the contraction property $n$ times (since $m > n$), yielding:

$$A \leq k^n \cdot d(\theta, T_{n+1} \cdots T_m(\theta)).$$

Now a simple application of the triangle inequality produces

$$
\begin{aligned}
A \leq \ & k^n \Big( d(\theta, T_{n+1}(\theta)) \\
& + d(T_{n+1}(\theta), T_{n+1}T_{n+2}(\theta)) + \cdots \\
& \cdots + d(T_{n+1} \cdots T_{m-1}(\theta), T_{n+1} \cdots T_m(\theta)) \Big).
\end{aligned}
$$

At this point, we can use the condition in (2) for the first term $d(\theta, T_{n+1}(\theta)) < D$ and in conjunction with the contraction property for the subsequent terms in the sum, yielding:

$$
\begin{aligned}
A & \leq k^n \left( D + D \cdot k + \cdots + D \cdot k^{m-n+1} \right) \\
& \leq D \cdot k^n \cdot \left( \frac{1 - k^{n-m+2}}{1 - k} \right) \\
& \leq \frac{D \cdot k^n}{1 - k},
\end{aligned}
$$

where we used the sum of a geometric series. Now, this yields that $A \to 0$ as $n \to \infty$, meaning that the sequence $\theta_n$ is a Cauchy sequence and thus converges to a certain point $\theta^*$ since $\Omega$ is complete. Then, taking the limit $m \to \infty$ we obtain

$$d(\theta_n, \theta^*) \leq \frac{D \cdot k^n}{1 - k} = C \cdot k^n,$$

with $C := \frac{D}{1-k}$, which shows the exponential convergence rate. $\qquad\square$

*Remark* 2.3. The contractivity assumption for the operators $T_i$'s is in general too restrictive for deep-learning settings. However, the proof of Theorem 2.2 still works in a more generalized setting that encompasses the non-convexity of deep-learning loss-landscapes. The price to pay though is the introduction of the more abstract notion of a *pseudo-metrics* for which the argument in the proof of Theorem 2.2 still works verbatim. We give the details of this more complicated approach in Appendix G.

As for the condition in (2), let us verify that it is satisfied for SGD in the case of an overparametrized model like a neural network. Namely, in this setting, we generally consider losses of the form $L(\theta) = \frac{1}{N} \sum_{i=1}^{N} L_i(\theta)$, where $L_i$ is the loss computed on batch $B_i$ with $L_i(\theta)$ being non-negative. Because of the over-parameterization, there generally exists points $\theta^*$ where the full loss vanishes. This in turns implies that $L_i(\theta^*) = 0$ for all $i$. For SGD, the operators are $T_i(\theta) = \theta - h\nabla L_i(\theta)$. Now it is easy to see that (2) is satisfied: $d(\theta^*, T_i(\theta^*)) = \|\theta^* - \theta^* - h\nabla L_i(\theta^*)\| = 0$, since the global zero $\theta^*$ is a critical point for all the $L_i$'s.

Now here is a major point: The contraction mapping principle fails with the forward order. As the next example demonstrates, it is not enough for the operator $T_i$'s to be uniform contractions for the forward sequence of iterates to converge, while this is always true for the backward sequence because of Theorem 2.2.

*Example* 2.4. (**Forward iterations counter-example.**) Here is an extreme example illustrating the convergence failure for the forward sequence of iterates (even when the maps are uniform contractions), while the backward sequence converges to a single point under the conditions of Theorem 2.2. Consider the two constant maps $S(\theta) = x_0$, and $U(\theta) = y_0$, which are contractions with $k = 0$. Assume that $x_0 \neq y_0$, now also assume that each $T_i$ is either equal to $S$ or $U$ with equal probability to be selected. Then, independently of $\theta$, the forward iterates

$$T_n T_{n-1} \cdots T_1(\theta)$$

will jump between $x_0$ and $y_0$, according to whether $T_n$ is $S$ or $U$ at that particular $n$. Thus, no convergence to a single point is possible, although the sequence converges to a probability uniformly distributed in the two outcomes, since for each forward iterate either outcome has probability 1/2. On the contrary, the backward trajectory

$$T_1 T_2 \cdots T_n(\theta)$$

will always converge to a single point determined by the first element in the sequence: either to $x_0$ if $T_1 = S$ or to $y_0$ if $T_1 = U$.

## 2.1 A fundamental example of a contraction

We now describe how the SGD updates are uniform contractions when the batch losses are strongly convex near a minimum. First, consider a strongly convex smooth function $L : \mathbb{R}^d \to \mathbb{R}$. Recall that $L$ is *strongly convex* if there is a constant $m > 0$ such that for all $\theta_1, \theta_2 \in \mathbb{R}^d$

$$\langle \nabla L(\theta_1) - \nabla L(\theta_2), \theta_1 - \theta_2 \rangle \geq m \left\| \theta_1 - \theta_2 \right\|^2, \tag{3}$$

where $\langle a, b \rangle = a^T b$ is the inner product between two vectors. Furthermore, we assume that there is a constant $M > m$ such that

$$\left\| \nabla L(\theta_1) - \nabla L(\theta_2) \right\| \leq M \left\| \theta_1 - \theta_2 \right\|. \tag{4}$$

This is sometimes denoted as $L \in \mathcal{S}_{m,M}^{1,1}(\mathbb{R}^d)$, see Nesterov (2013). The following lemma (whose proof is in Appendix F) shows that for strongly convex functions we can always find a learning rate small enough so that the corresponding gradient descent update is a contraction.

**Lemma 2.5.** *Let $L(\theta)$ satisfy* (3) *and* (4) *and define the map $T(\theta) = \theta - h\nabla L(\theta)$. Then*

$$\left\| T(\theta_1) - T(\theta_2) \right\| \leq \sqrt{1 - 2hm + h^2 M^2} \left\| \theta_1 - \theta_2 \right\|.$$

*In particular, for small enough $h \in (0, 1)$ (depending on $m$ and $M$) the map $T$ is a uniform contraction.*

Now if we suppose that batch losses $L_i$'s are strongly convex (possibly near global minima), then Lemma 2.5 tells us that the SGD maps $T_i(\theta) = \theta - h\nabla L_i(\theta)$ are uniform contractions. Thus, we can apply Theorem 2.2 and see that the sequence of backward iterates, $\theta_n = T_1 T_2 \cdots T_n(\theta)$, converges to a point.

## 2.2 Relation between forward, backward, and generalization

In this section we address the question of the generalization power of backward iterates, which we argue is the same as that of the forward iterates. The relationship with generalization is implicit through Theorem 2.6 below which states that backward convergence points are distributed according to the forward stationary distribution. It follows that the best test-performance forward-solutions can be reached by the backward trajectories with the same probability. This a good thing as it shows that the backward trajectories are more stable but without test-performance reduction on average, which is often not the case as for instance the increased stability obtained by increasing the learning rate or decreasing the batch-size (see Dherin et al. (2022)). In particular, the bias of backward SGD is statistically equivalent to that of standard SGD. Consequently, the convergence point of the backward iterates cannot outperform the best solutions attainable by the forward iterates.

The goal of this work aims to demonstrate that the composition order creates two different processes, which we could described intuitively as follows:

- The forward iterate process converges toward a distribution, "forgetting" the initial batches and being most influenced by the recent ones. This makes this process robust to a bad start but leaves it perpetually noisy.

- The backward iterate process converges toward a point, "remembering" the initial batches but "forgetting" the recent stochastic noise.

Theorem 2.6 formalizes this intuition and essentially generalizes the situation already seen in Example 2.4, an actual realization of a forward trajectory converges to a distribution which is uniformly distributed between the two outcomes. On the other hand, any realization of the backward trajectory always converges to a point sampled from this distribution. Let's formalize this for the general case.

First of all, recall that a map $T : \Omega \to \Omega$ produces a corresponding push-forward map $T_* : P(\Omega) \to P(\Omega)$ at the level of probability measures $P(\Omega)$ on $\Omega$. Recall that a probability measure $\mu \in P(\Omega)$ associates to each subset $A \subset \Omega$ a number $\mu(A)$ modelling the probability $\text{Prob}_\mu(\theta \in A)$ of $\theta$ being in the subset $A$ if $\theta$ is sampled from $\mu$. Now, if $\mu$ is a probability measure on $\Omega$ then the push forward $T_*\mu(A) = \mu(T^{-1}(A))$ for $A \subset \Omega$ computes the probability $\text{Prob}_\mu(T(\theta) \in A)$ of $T(\theta)$ being in $A$ if $\theta$ is sampled according to $\mu$.

Therefore, the forward sequence $(T_n \cdots T_2 T_1) : \Omega \to \Omega$ induces a push-forward $(T_n \cdots T_2 T_1)_*$ that maps an initial distribution $\mu_0$ to the probability measure $\mu_n = (T_n \cdots T_2 T_1)_* \mu_0$ modeling the probability distribution of the $n^{th}$ forward iterate $\theta_n = T_n \cdots T_2 T_1(\theta_0)$ when the initial point $\theta_0$ is randomly sampled from $\mu_0$. Observe that the forward iterates $\theta_n$ form a sequence of random variables, which is a Markov chain (since for the forward iterates we have that $\theta_n = T_n(\theta_{n-1})$). Many previous works (see related work in Section 1.1) have shown under different assumptions that this sequence of forward iterates do not converge to a point when using fixed learning rates, but rather their probability distributions $\mu_n$ converge to a limiting probability distribution, called the stationary distribution of the Markov chain. In other words, starting at initial point $\theta_0$ the distribution $\mu_n = (T_n)_* \cdots (T_1)_* \delta_{\theta_0}$ of the forward iterate $\theta_n$ converges to a stationary distribution $\mu_{\theta^*}$. Recall that $\delta_{\theta_0}$ is the Dirac measure concentrated at $\theta_0$ (i.e., $\delta_{\theta_0}(A)$ is 1 if $A$ contains $\theta_0$ and 0 otherwise).

On the contrary, as we have seen in Theorem 2.2, the backward iterates converge to actual points in regions where the maps $T_i$'s become contractions (which we will call *contractive regions*). The following theorem (whose proof is in Appendix E) shows the relation between the point-wise convergence of the backward iterates and the distributional convergence of the forward iterates:

**Theorem 2.6.** *Consider a sequence $\{T_i\}$, $i = 1, 2, \ldots$ of independent and identically distributed random operators. Suppose that for $\theta_0 \in \Omega = \mathbb{R}^d$ the backward iterates converge to a random point (randomness is due to the sampling of the random operators $T_i$'s):*

$$T_1 T_2 \cdots T_n(\theta_0) \longrightarrow \theta^* \quad as \quad n \to \infty.$$

*Then the probability distribution of the forward iterates from $\theta_0$ converge (in distribution) to a stationary probability measure $\mu_{\theta^*}$. Moreover, the random point $\theta^*$ is distributed according to the same forward iterate stationary distribution $\mu_{\theta^*}$.*

## 3 Two explicit examples on stochastic gradient descent

We now present two examples that provide an intuition on why backward SGD converges to a point rather than to a distribution as forward SGD does, when the learning rate is fixed. The reason in both examples is that the stochastic noise added to the parameter iterates at each step due to the randomization of the batch diminishes to zero in backward SGD as the training progress even with fixed learning rates, while this stochastic noise stays unchanged at each step of forward SGD, requiring a learning rate decay for convergence.

### 3.1 Example: Quadratic loss

Consider the quadratic loss function $L(\theta) = \theta^2/2$ with $\theta \in \mathbb{R}$. Its gradient is $\theta$. We want to find the minimum of $L$ (which of course is 0 at $\theta = 0$) by a stochastic gradient descent. To model the batch noise, we add an i.i.d random error $\epsilon_i$ to the gradient at step $i$, yielding the iteration procedure:

$$T_i(\theta) = \theta - h\Big(\nabla L(\theta) - \epsilon_i\Big) = (1-h)\theta + h\epsilon_i,$$

where $h$ is the learning rate. These maps are uniform contractions provided that $\theta \in (0, 1)$ since

$$|T_i(\theta_1) - T_i(\theta_2)| = |(1-h)(\theta_1 - \theta_2)| \leq |1-h||\theta_1 - \theta_2|.$$

Now we would like to compare the behavior of the forward trajectory to that of the backward trajectory for the $T_i$'s. In order to do this we begin by calculating the forward iterates:

$$
\begin{aligned}
T_2 T_1(\theta) &= (1-h)((1-h)\theta + h\epsilon_1) + h\epsilon_2 \\
&= (1-h)^2\theta + h\epsilon_2 + (1-h)h\epsilon_1.
\end{aligned}
$$

One sees easily how this continues:

$$T_n \cdots T_1(\theta) = (1-h)^n \theta + h\epsilon_n + \cdots + h(1-h)^{n-1}\epsilon_1.$$

By switching indices, for the backward iterates we obtain:

$$T_1 \cdots T_n(\theta) = (1-h)^n \theta + h\epsilon_1 + \cdots + h(1-h)^{n-1}\epsilon_n.$$

We see that the forward iterates receive at each step $n$ an additional error $h\epsilon_n$ which stays constant during the whole trajectory preventing point-convergence, while for backward iterates that same error $h(1-h)^{n-1}\epsilon_n$ is decaying to zero as the trajectory progresses.

### 3.2 Example: Learning dynamics close to minima

The previous example was a warm up for the following more realistic example. However, the key idea is identical. Consider a loss function

$$L(\theta) = \frac{1}{N}\sum_{i=1}^{N} L_i(\theta),$$

where $L_i$ is the loss computed on the random batch $B_i$. The SGD update is

$$T_i(\theta) = \theta - h\nabla L_i(\theta)$$

where $h$ is again the learning rate. We now compare forward and backward SGD dynamics sufficiently close to a minimum $\theta^*$ of the loss function so that we can approximate well the full loss gradient $g(\theta) = \nabla L(\theta)$ using the loss Hessian $H(\theta) = \nabla g(\theta)$: i.e., a first order Taylor's expansion of the gradient at the minimum yields

$$g(\theta) \simeq g(\theta^*) + H(\theta^*)(\theta - \theta^*) = H(\theta^*)(\theta - \theta^*), \tag{5}$$

since at a minimum $g(\theta^*) = 0$. To simplify the notation, we will denote by $g_i(\theta)$ the gradient of the batch loss $L_i(\theta)$, which is now a random vector (because the batch is random). We can model this stochastic gradient as follows: $g_i(x) = g(x) - \epsilon_i$, where we assume that the differences $\epsilon_i = g(x) - g_i(x)$ are i.i.d random vectors with zero mean. Using gradient approximation (5) we have that a single gradient update is given in a form very similar to that of the previous example:

$$T_i(\theta) = \theta - h\Big(g(\theta) - \epsilon_i\Big) \simeq \theta^* + (1-hH)(\theta - \theta^*) + h\epsilon_i$$

where $H$ denotes the Hessian evaluated at the minimum $\theta^*$. Since

$$\|T_i(\theta_1) - T_i(\theta_2)\| \le \|1 - hH\|\|\theta_1 - \theta_2\|$$

and since $\|1 - hH\| < 1$ for $h$ strictly smaller than $2/\lambda_{\max}$ (as in Example 2.1), Theorem 2.2 tells us that the backward trajectory converges. Now let us have a look at the form of the forward and backward iterates in this particular case to give us an intuition about why this is true. In a similar way than in the previous example, the $n^{th}$ forward SGD iterate is given by

$$T_n \cdots T_1(\theta) = \theta^* + (1-hH)^n(\theta - \theta^*) + h\epsilon_n + h(1-hH)\epsilon_{n-1} + \cdots \cdots + h(1-hH)^{n-1}\epsilon_1.$$

While the term $(1-hH)^n(\theta - \theta^*)$ converges to zero (when $h < 2/\lambda_{max}$), we see that at each step $n$ a fresh random vector $h\epsilon_n$ is added to the iterate, making the whole sequence converge to a distribution rather than to a point. On the other hand the backward iterate is obtained by reversing the indices:

$$T_1 \cdots T_n(\theta) = \theta^* + (1-hH)^n(\theta - \theta^*) + h\epsilon_1 + h(1-hH)\epsilon_2 + \cdots \cdots + h(1-hH)^{n-1}\epsilon_n.$$

This expression converges as $n \to \infty$ since the terms are decreasing at an exponential rate as $n$ grows. Thus the iterates converge to a point now rather than a distribution. Looking at the actual form of the backward iterates, it is clear that they are distributed by the probability measure given by the forward iterates as prescribed by Theorem 2.6.

## 4    Experiments

**Note on plotting multiple seeds:** We are interested in the variability per realization of the backward and forward trajectories. The backward trajectories are more stable individually along their own path, but these paths can be very different from seed to seed (because of the convergence toward different points). Therefore the phenomenon is much clearer when the seeds are plotted individually rather than averaged, which creates artificially more variability in the backward trajectories as there really is on each individual realization. This is why we are reporting the various seed plots individually and not as a single averaged plot with error bars. Note that the increased stability and point convergence is visible in each of the seeds, which are added to the Appendix B.

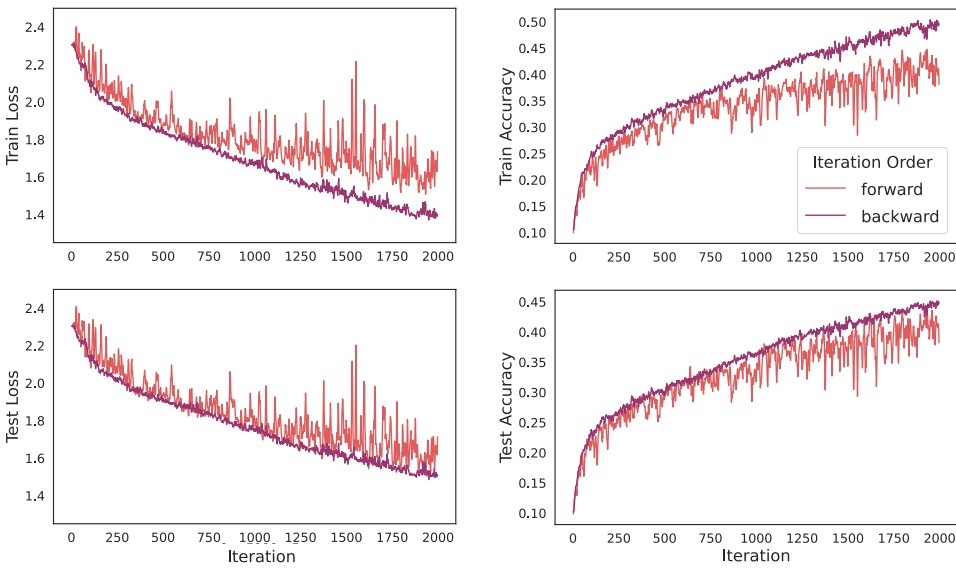

Figure 2: Backward SGD exhibits decreased variance and increased stability compared to forward SGD for a ResNet-18 model trained on CIFAR-10. The additional seeds are in Appendix B.1.

**Increased convergence stability for backward SGD:** We trained a ResNet-18 with stochastic gradient descent and no regularization on the CIFAR-10 dataset Krizhevsky (2009). We used a learning rate of 0.025 and a batch-size of 8. In Figure 2, we recorded the learning curves at *each* gradient update for both the forward and backward iterations for 2000 steps. The additional seeds are in Appendix B.1. We observe that the training loss for the backward trajectory is more stable, converges faster, and has less variability than the forward trajectory, and similarly for all the other learning curves. In Appendix A, we repeat the experiment with different datasets (synthetic dataset, FashionMNIST, and CIFAR-100) as well as different architectures (ResNet-50, VGG, and MLP) and using different base optimizers (AdamW). Each time we observe the same phenomenon for all 5 seeds.

**Convergence toward points v.s. distributions:** We trained a MLP with 5 layers of 500 neurons each with stochastic gradient descent with no regularization to learn FashionMNIST Xiao et al. (2017). The batch-size was set to 8 while the learning rate was 0.001. In Figure 3, we report the test accuracy at every single step for 2000 steps. At step 1000, we reset the point from which we perform backward SGD to that of the parameter at this step. As a result we see that the backward trajectory from that reset point seems to converge again, but to a different point (with different test accuracy), while forward SGD seems to oscillate between these two points. This is in line with the theoretical prediction that backward trajectories SGD converge toward points distributed according to the distribution induced by forward trajectories (see Theorem 2.6). The additional seeds are in Appendix B.2.

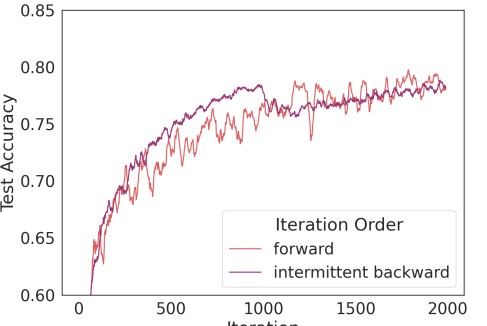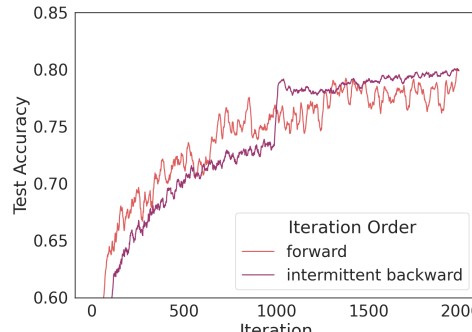

Figure 3: Backward SGD converges toward a different minimum after resetting the initialization point at step 1000 ("intermittent backward") while forward SGD oscillates between them for MLP trained on FashionMNIST. **Top:** On the first seed, backward changes from a higher test-performance trajectory to a lower test-performance trajectory at the reset step 1000. **Bottom:** On the second seed, backward changes this time from a trajectory converging to a lower test-performance point to a trajectory converging to a higher test-performance point. The other seeds can be found in Appendix B.2 including all the learning curves.

## 5    Conclusion and future work

In this paper, we analyzed the impact of two different composition orders (to produce each iterate) on the convergence and stability of stochastic learning trajectories. Although the very idea of backward trajectories may seem strange and counterintuitive at first, we show that this approach of backward optimization has clear advantages. Namely, we show theoretically that the backward trajectories converge toward actual points while the forward trajectories converge toward probability distributions in regions where the optimization maps become contractions, leading to improved stability and convergence. We observed this phenomenon experimentally on a number of datasets (synthetic, FashionMNIST, CIFAR-10, and CIFAR-100) and neural network architectures (MLP, VGG-19, ResNet-18, and ResNet-50), making it relevant to deep learning. This phenomenon points toward the fact that the ordering in which the data examples are consumed to produce each parameter iterate impacts the properties of both the learning trajectories and their convergence points. In particular the last examples used to produce a parameter iterate seem to have great importance. Indeed, the backward order is the only order in which the sequence of the last examples used to produce a parameter iterate remains consistent at each training step. As we saw, this yields convergence toward a point rather than oscillations between solutions of varied performances.

However, realistic applications of the full backward order are challenging due to the prohibitive computation time that grows quadratically with the number of batches consumed. We now point toward conjectural workarounds and applications to showcase possible exploitation of this phenomenon. This is outside of the scope of this paper; nevertheless, we give supporting evidence in the appendix when possible.

One possible strategy to mitigate the intensive computational requirements of the full backward order is to apply it only partially on a fixed window of iterations from a point that is reset periodically. This way the extra computation time is capped by the window size. We see that this approach leads to convergence and increased stability in Figure 3; however the backward trajectory may change and converge to a different point at each reset. We can imagine strategies to initiate this reset only when the performance of the new backward trajectory increases, leading to stable convergence to higher test performance points. Another possibility is to start the backward order only later in training to force convergence toward a single solution after the forward trajectory has done enough exploration of the space as demonstrated in Figure 11 in Appendix A. We imagine that this approach may possibly lead toward more reliable early-stopping criteria. Similarly, application of the backward order at the very beginning of training may give an accurate idea of whether a

specific hyperparameter setting is promising or needs to be aborted by creating stabler trajectories whose learning curves are easier to interpret early on.

An orthogonal approach to deal with the computational challenges of backward optimization is to approximate the backward trajectory. In Appendix C, we show how to compute a backward iterate from a cheaper forward iterate to which we add a correction that can be computed completely independently. We compute these corrections up to order 2 in Theorem C.2 Theorem C.2, namely:

$$T_1 \cdots T_n = T_n \cdots T_1 + h^2 \sum_{1 \leq i < j \leq n} [\nabla L_i, \nabla L_j] + \mathcal{O}(h^3)$$

where $[\nabla L_i, \nabla L_j](\theta) = H_i(\theta)\nabla L_j(\theta) - H_j(\theta)\nabla L_i(\theta)$ is the Lie bracket between the vector fields $\nabla L_i$ and $\nabla L_j$. Even though the approximation we provide is not immediately useful because the higher order terms seem to matter, it probably will be useful if higher order corrections are included. As a proof of the usefulness of these type of corrections added to the forward iterates to emulate alternate orderings, we show in Appendix D that the corrections we computed at second order are already enough to approximate an average between all the possible iteration orders on a batch window. Adding these corrections to the forward iterates produces a beneficial regularizer that mimics small batch training regularization. In fact, a training step can be seen as a parameter average of mixture of models where each model is trained not from a different seed but from a different batch order. We hope that this paper will bring awareness and foster more research toward understanding and exploiting the role of iteration order in the production of models with more consistent properties along more stable learning trajectories.

## Broader Impact Statement

This work focused on the theoretical aspect of learning algorithms, especially stability. We do not foresee any negative impact from this theoretical work.

## Acknowledgments

The third-listed author was supported by the Swiss NSF Grants 200020-200400 and 200021-212864, and by the Swedish Research Council Grant 104651320. We are grateful to Massimo Horvath for several insightful comments on an earlier version of the manuscript. We would like to thank Michael Wunder, Peter Bartlett, George Dahl, Atish Agarwala, and Scott Yak for helpful discussion and feedback.

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

# A    Additional Experiments

**Note on plotting multiple seeds:** We are interested in the variability per realization of the backward and forward trajectories. The backward trajectories are more stable individually along their own paths, but these paths can be very different from seed to seed (because of the convergence toward different points). Therefore, the phenomenon is much clearer when the seeds are plotted individually rather than averaged, which creates artificially more variability in the backward trajectories as there really is on each individual realization. This is why we are reporting the various seed plots individually and not as a single averaged plot with error bars. Note that the increased stability and point convergence is visible in each of the seeds.

## A.1 Regression on synthetic datasets

In this experiment, to verify the increased stability of backward SGD over the standard forward version, we trained with both algorithms a neural network with 3 layers of 300 neurons with no regularization each to regress 100 points uniformly sampled from function graphs $D = \{(x_i, y_i) : y_i = f(x_i) \text{ with } x_i = -1 + \frac{2i}{100}, i = 0, \ldots, 100\}$ using the functions $f(x) = x^2$ (Figure 4), $f(x) = \cos(10x)$ (Figure 5), and $f(x) = x^3$ (Figure 6). In all cases, we can observe a much higher stability for backward SGD over the forward version.

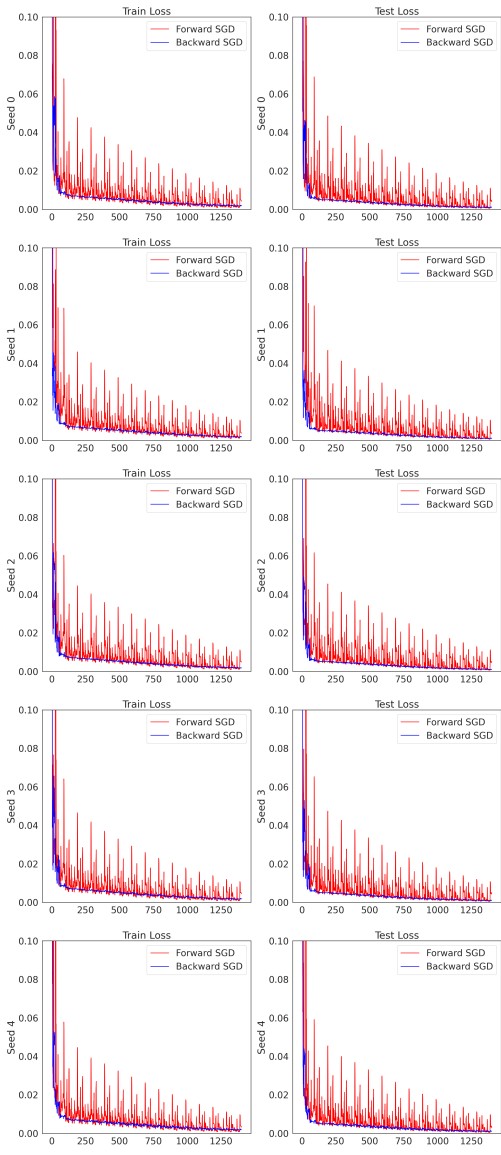

Figure 4: Decreased variance and increased stability in train (left) and test (right) losses for backward SGD compared to forward SGD for all 5 seeds. The data was sampled from $f(x) = x^2$ and the training performed with batch size 1 and learning rate 0.05 for 1400 steps.

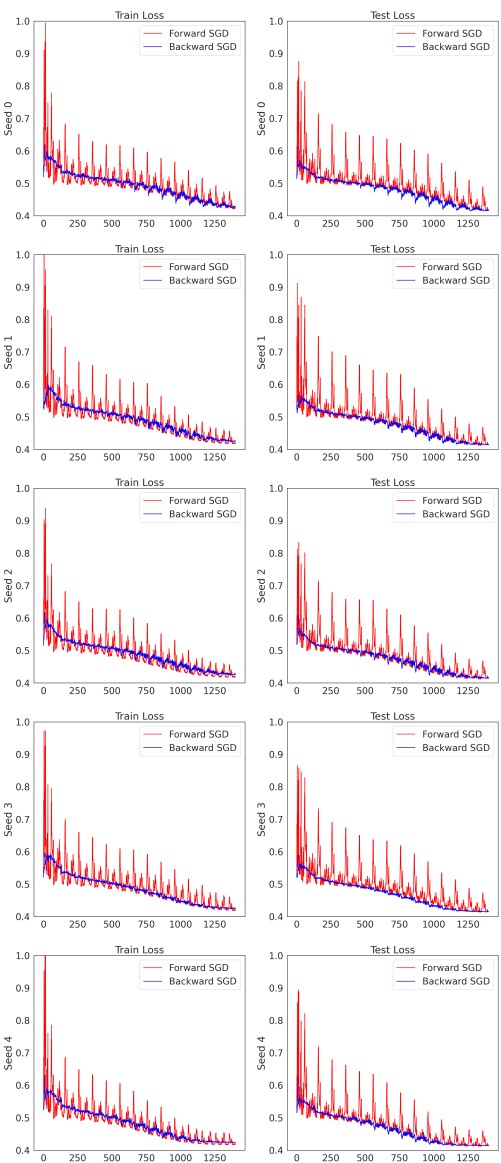

Figure 5: Decreased variance and increased stability in train (left) and test (right) losses for backward SGD compared to forward SGD for all 5 seeds. The data was sampled from $f(x) = \cos(10x)$ and the training performed with batch size 1 and learning rate 0.02 for 1400 steps.

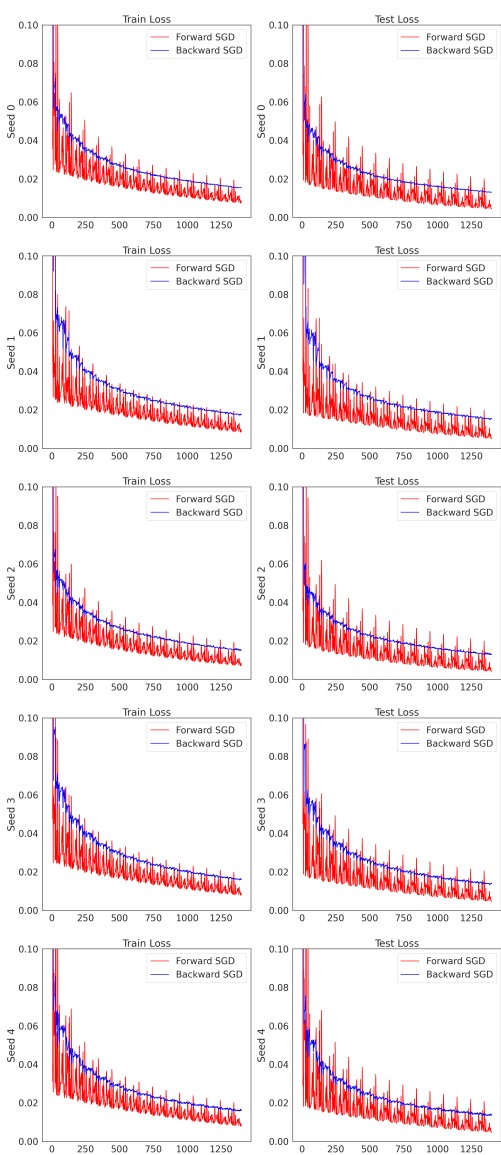

Figure 6: Decreased variance and increased stability in train (left) and test (right) losses for backward SGD compared to forward SGD for all 5 seeds. The data was sampled from $f(x) = x^3$ and the training performed with batch size 1 and learning rate 0.02 for 1400 steps.

## A.2 MLP trained on Fashion-MNIST

In this experiment, to verify the increased stability of backward SGD over the standard forward version, we trained a MLP with 5 layers of 500 neurons each with no regularization using both forward and backward stochastic gradient descent with no regularization on the Fashion-MNIST dataset Xiao et al. (2017). We repeated the experiment for 5 different random seeds. For all seeds, we used a learning rate of 0.001 and a batch size of 8. In Figure 7, we recorded the training loss at each gradient update for both the forward end backward iteration and plotted the 5 seeds separately (one seed for each row). For each seed, we observe that the training loss for the backward iterations is more stable and converges faster with less variability when compared to the forward iterations. We also observe stability improvements for all other learning curves for the backward trajectories.

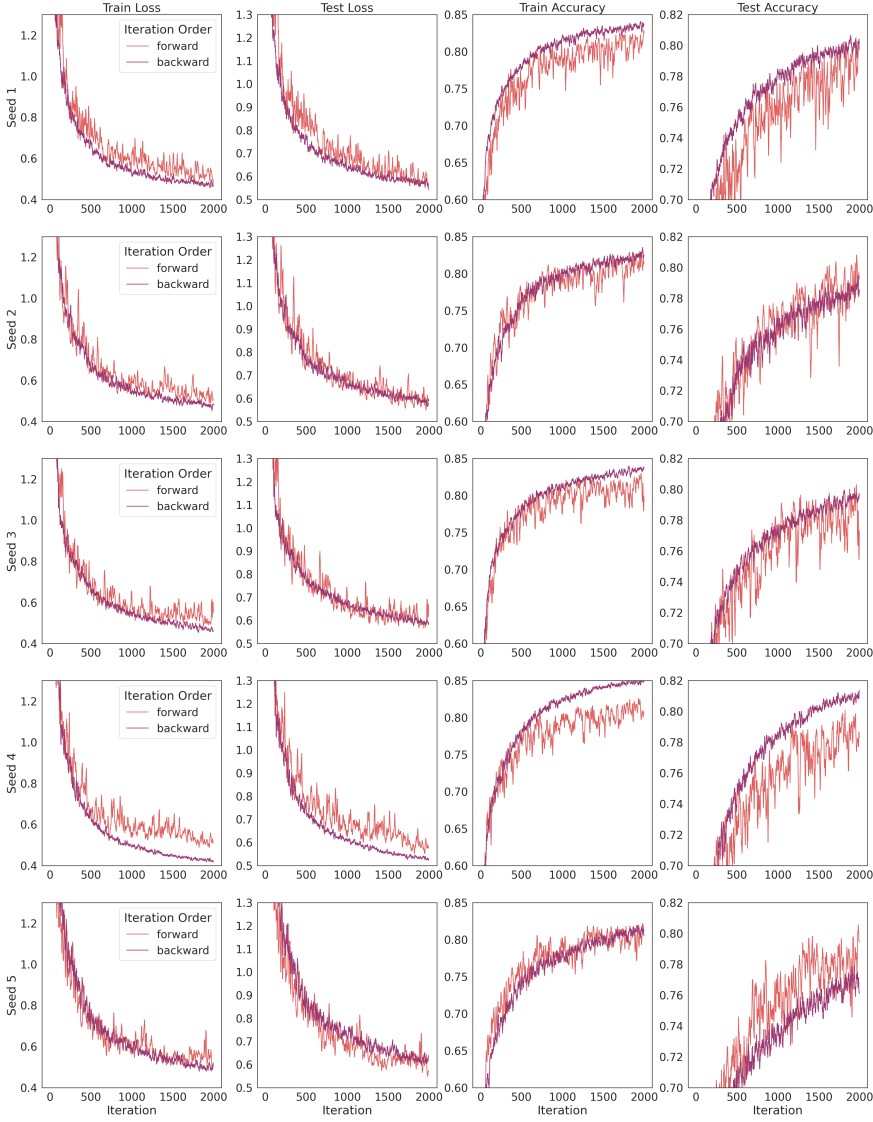

Figure 7: MLP trained on Fashion-MNIST. Training with backward SGD significantly reduces the variance and increases the stability of both the train and test loss as well as the train and test and accuracy compared to forward SGD. This behavior is consistent across all seeds.

## A.3 VGG-19 trained on CIFAR-10

In this experiment, to verify the increased stability of backward SGD over the standard forward version, we trained a VGG-19 model Simonyan & Zisserman (2014) using both forward and backward stochastic gradient descent with no regularization on the CIFAR-10 dataset Krizhevsky (2009). We repeated the experiment for 5 different random seeds. In all seeds, we used a learning rate of 0.001 and a batch-size of 8. In Figure 8, we plot the training and test loss at each gradient update for both the forward and backward iteration. We plotted the 5 seeds separately, represented by each row. Note that for each seed, we observe that the training loss for the backward iterations is again more stable, converges faster, and has less variability than the forward iterations.

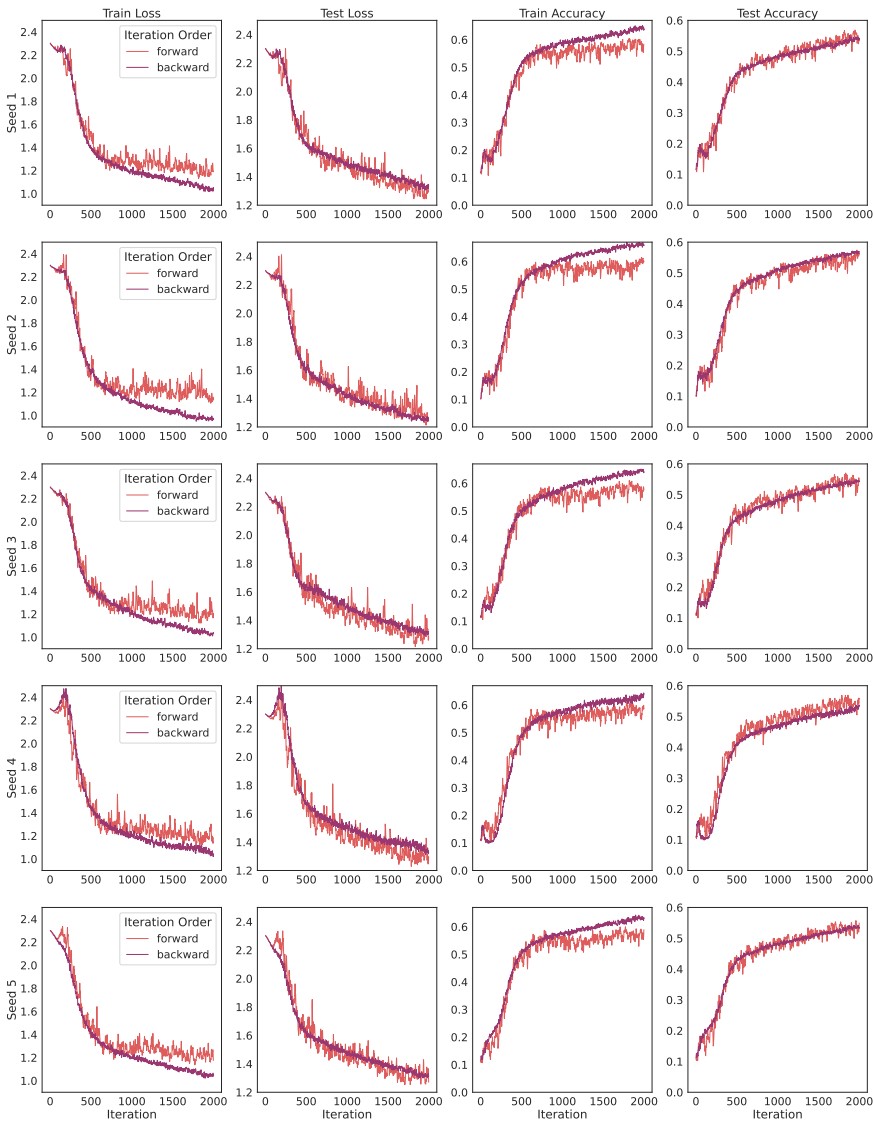

Figure 8: VGG-19 trained on CIFAR-10. Training with backward SGD significantly reduces the variance and increases the stability of both the train and test loss as well as the train and test and accuracy compared to forward SGD. This behavior is consistent across all seeds.

## A.4 ResNet-50 trained on CIFAR-100

In this experiment, to verify the increased stability of backward SGD over the standard forward version, we trained a ResNet-50 model He et al. (2016) using both forward and backward stochastic gradient descent with no regularization on the CIFAR-100 dataset Krizhevsky (2009). We repeated the experiment for 5 different random seeds. In all seeds, we used a learning rate of 0.001 and a batch-size of 16. In Figure 9, we plot the training and test loss at each gradient update for both the forward and backward iteration. We plotted the 5 seeds separately, represented by each row. Note that for each seed, we observe that the training loss for the backward iterations is again more stable, converges faster, and has less variability than the forward iterations.

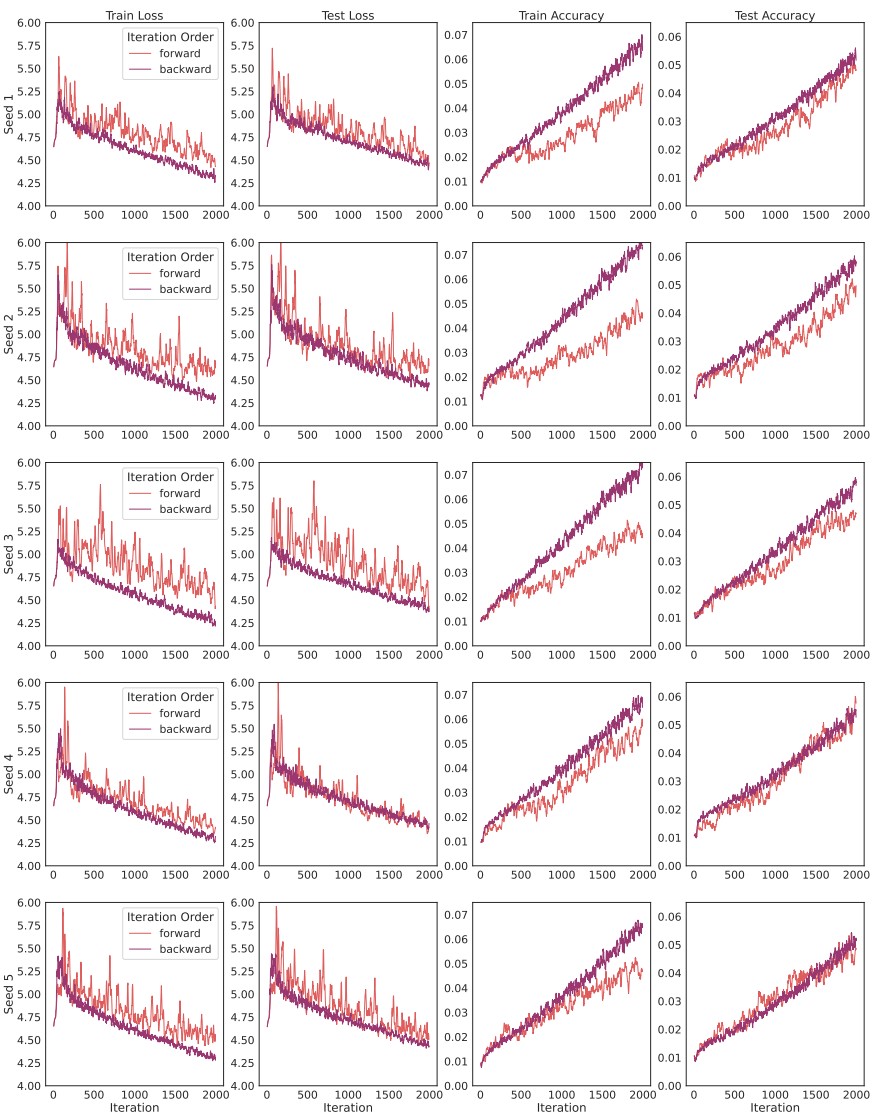

Figure 9: ResNet-50 trained on CIFAR-100. Training with backward SGD significantly reduces the variance and increases the stability of both the train and test loss as well as the train and test and accuracy compared to forward SGD. This behavior is consistent across all seeds.

## A.5 ResNet-18 trained on CIFAR-10 with AdamW

In this experiment we verify improved stability of backward iterates for a base optimizer other than vanilla SGD. AdamW Loshchilov & Hutter (2017) combines momentum from Adam with weight decay regularization and improves upon the standard Adam by preventing the adaptive learning rates from distorting the intended regularization strength.

Here, we use AdamW for both forward and backward gradient updates, only altering the order in which batches are consumed. We train a ResNet-18 model He et al. (2016) and do not apply any regularization on the CIFAR-10 dataset Krizhevsky (2009). We repeated the experiment for 5 different random seeds. In all seeds, we used a learning rate of 0.00025 and a batch-size of 8. In Figure 10, we plot the training and test loss at each gradient update for both the forward and backward iteration. We plotted the 5 seeds separately, represented by each row. Note that for each seed, we observe that the training loss for the backward iterations is again more stable, converges faster, and has less variability than the forward iterations.

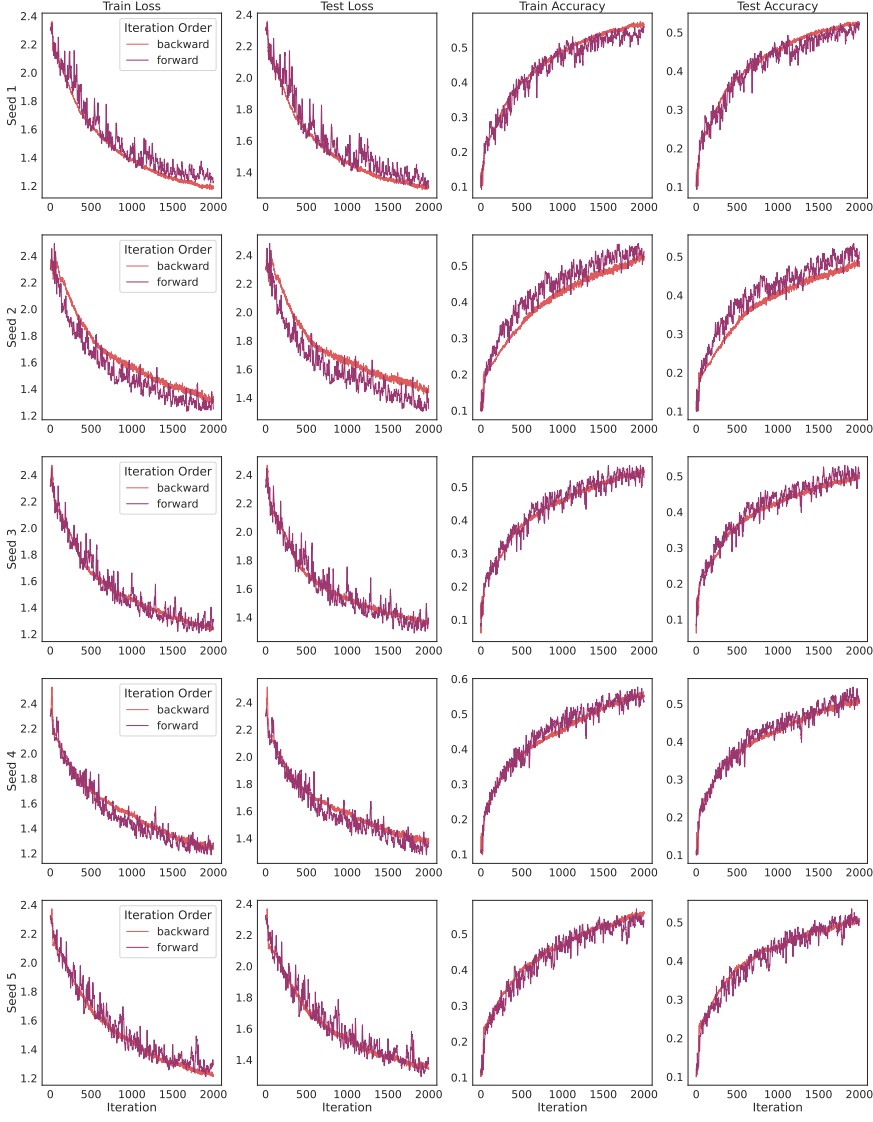

Figure 10: ResNet-18 trained on CIFAR-10 with AdamW. Training with backward AdamW significantly reduces the variance and increases the stability of both the train and test loss as well as the train and test and accuracy compared to forward AdamW. This behavior is consistent across all seeds.

## A.6 Backward stabilization after forward iterations

In this experiment, to verify the increased stability and convergence after turning on backward iterations after a number of forward iterations, we trained a MLP with 5 layers of 500 neurons each with stochastic gradient descent with no regularization to learn the 10 classes of Fashion-MNIST dataset Xiao et al. (2017). We repeated the experiments for 5 seeds. We used a learning rate of 0.001 and a batch-size of 8. In Figure 11, we recorded the learning curves at each gradient update for both the forward iteration and the backward iteration switched on after step 1000. Each seed is plotted independently. We observe that the training loss for the backward iterations has again more stable convergence once backward iterations are switched on after step 1000 than the forward iterations.

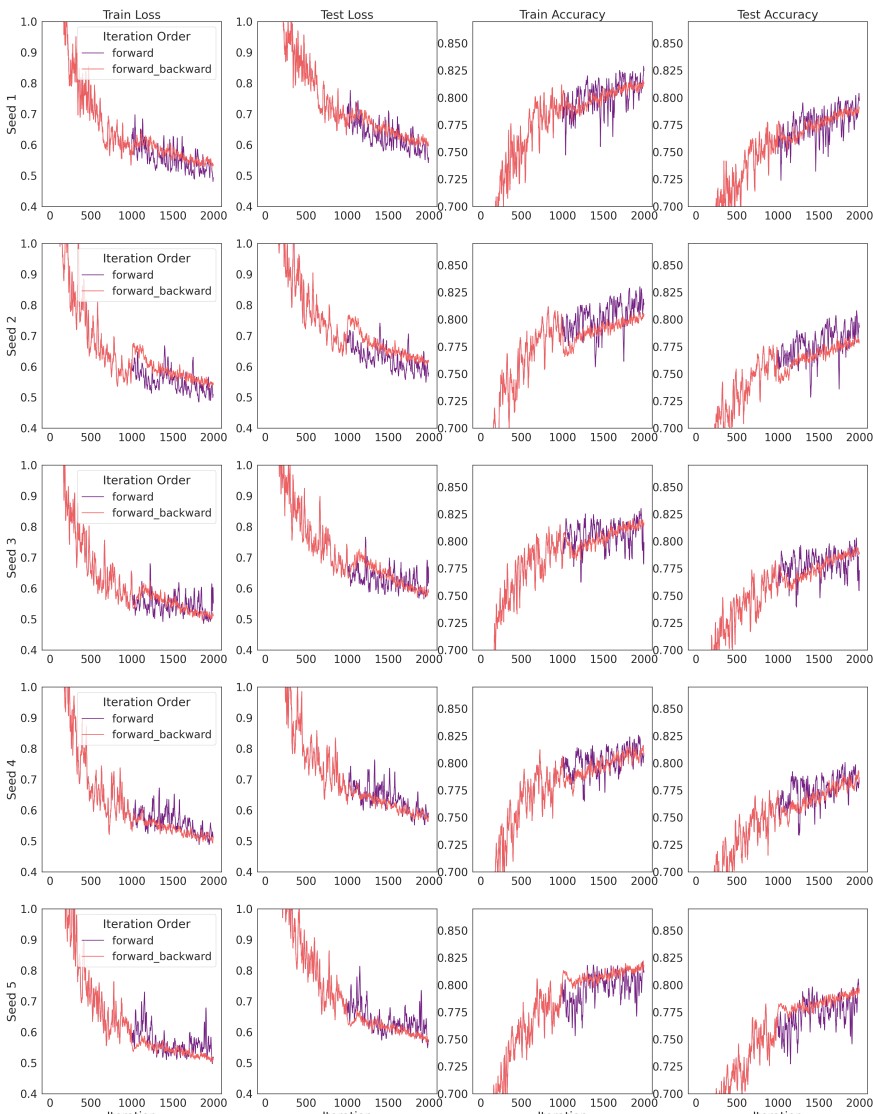

Figure 11: Decreased variance and increased stability for all learning curves and all seeds once backward SGD is switched on at step 1000 after forward SGD iterations for an MLP trained on Fashion MNIST.

### A.7 Continued stability of backward SGD throughout training.

In this experiment, to verify that the behavior of increased stability and convergence for backward iterates continues throughout training, we trained a ResNet-18 model on the CIFAR-10 dataset using the same hyperparameters as in Figure 2 (i.e., no regularization, a learning rate of 0.025 and batch size of 8) but training for 25000 steps instead of 2000. In Figure 12, we record the learning curves for both forward and backward SGD, performing model evaluation every 100 steps. Throughout training, we continue to notice that the training loss (as well as all the other learning curves) for the backward iterations is again more stable, converges faster, and has less variability than the forward iterations. Note, here we only performed this experiment for one seed because because of the high computational requirements for backward SGD with so many training steps.

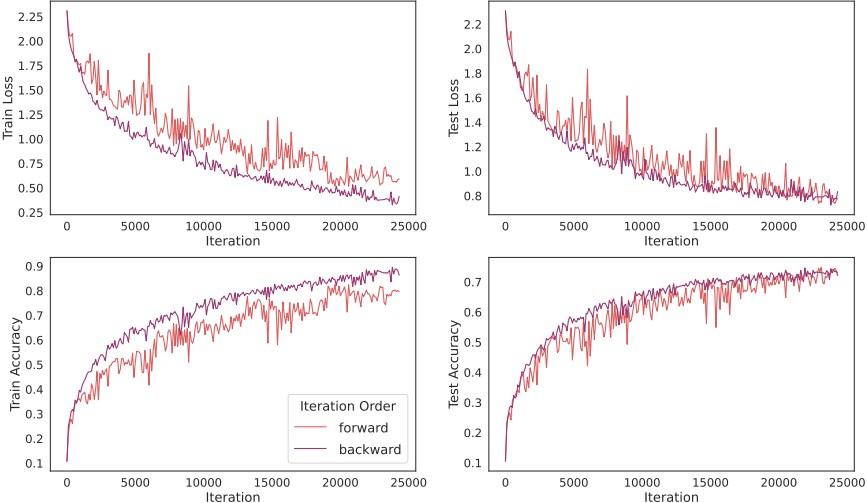

Figure 12: ResNet-18 trained on CIFAR-10 for 25000 steps. Decreased variance and increased stability throughout model training.

# B   Additional seeds to plots in the main paper

**Note on plotting multiple seeds:** We are interested in the variability per realization of the backward and forward trajectories. The backward trajectories are more stable individually along their own paths, but these paths can be very different from seed to seed (because of the convergence toward different points). Therefore the phenomenon is much clearer when the seeds are plotted individually rather than averaged, which creates artificially more variability in the backward trajectories as there really is on each individual realization. This is why we are reporting the various seed plots individually and not as a single averaged plot with error bars. Note that the increased stability and point convergence is visible in each of the seeds.

## B.1   All 5-seeds for ResNet-18 trained on CIFAR-10; cf. Figure 2

In Figure 13 below, we plot the learning curves for all five seeds as in Figure 2 of the main paper.

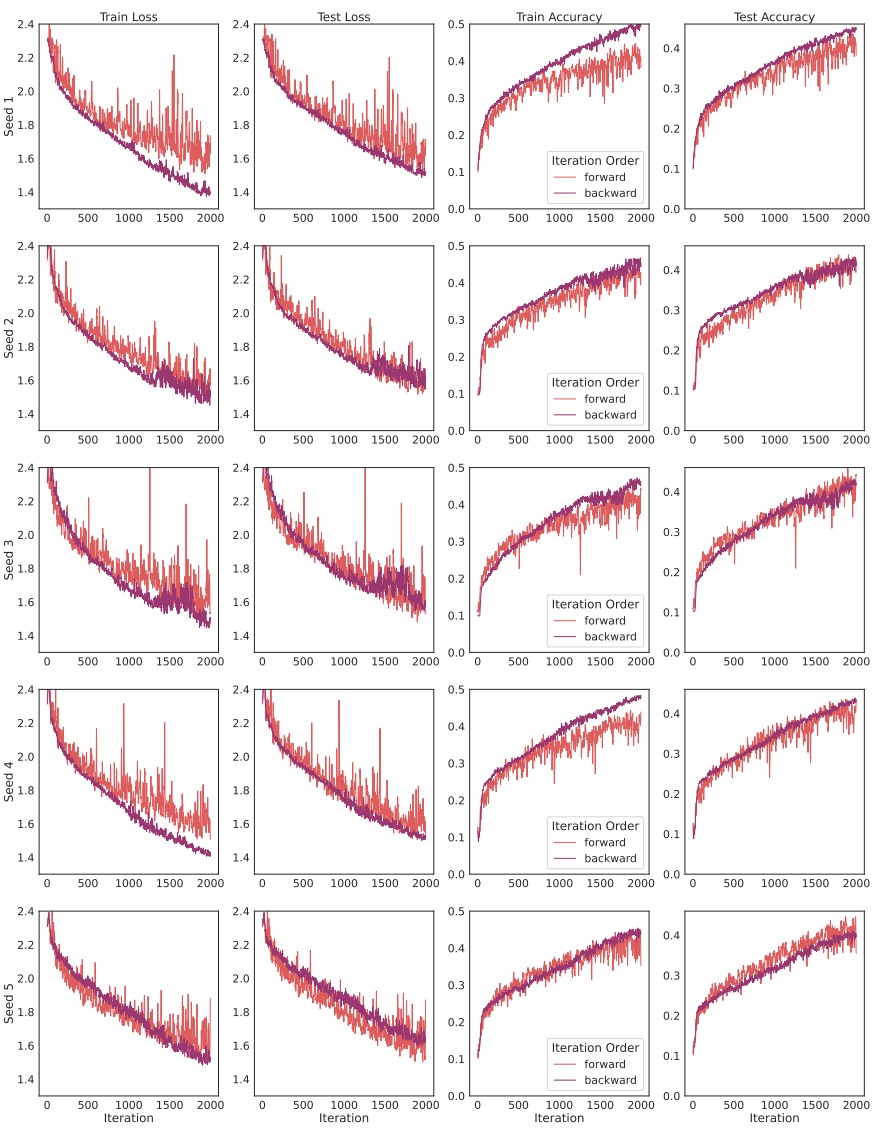

Figure 13:   All 5-seeds plot of Figure 2.

## B.2 All 5-seeds for MLP trained on FashionMNIST; cf. Figure 3

In Figure 14 below, we plot the learning curves for all five seeds as in Figure 3 of the main paper.

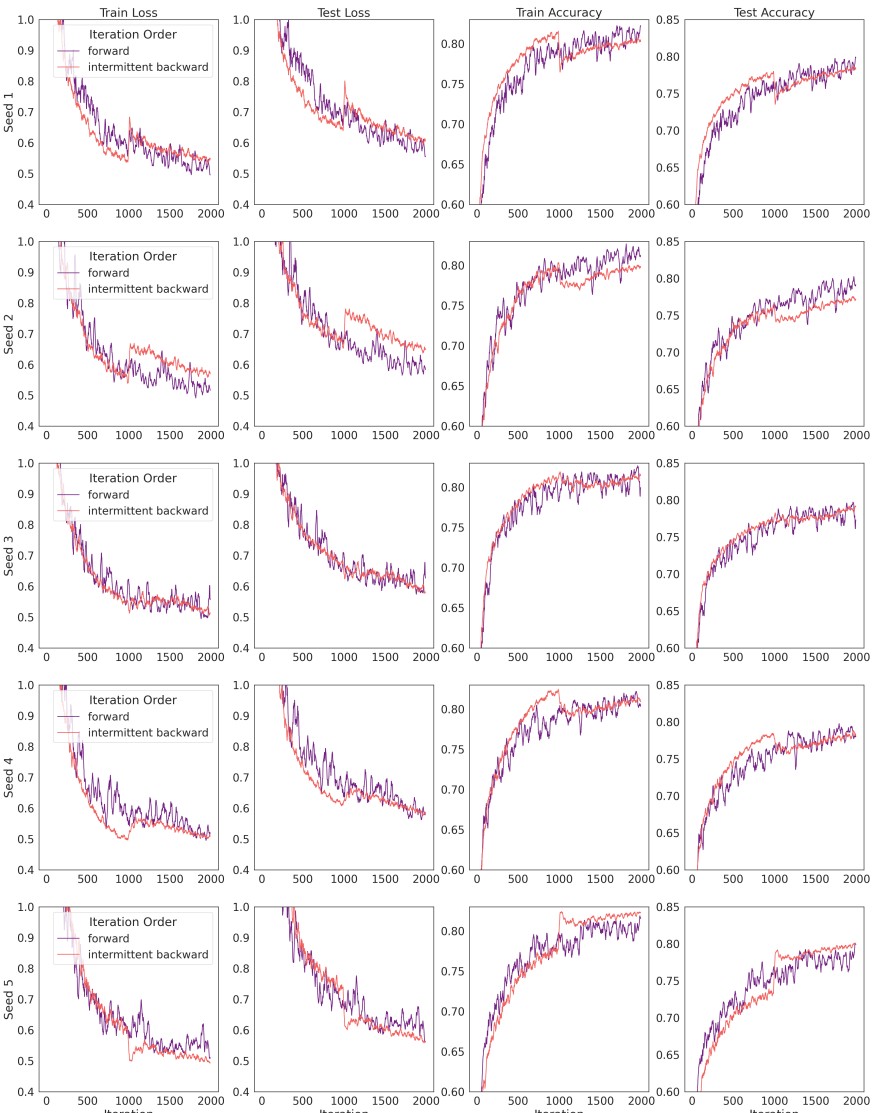

Figure 14: All 5-seeds plot of Figure 3

## C   Toward an approximate backward SGD

Backward SGD has increased stability and convergence over forward SGD but naive implementations are computationally intensive. In this section, we propose an approximation of the backward iterates by modifying the forward iterates with a Lie bracket term of order $\mathcal{O}(h^2)$ in the learning rate (Theorem C.2). This comes at an extra cost, but which can be made smaller than evaluating the backward iterates from the initial point at every step (Corollary C.4). Our approximation is valid

- for any optimizer of the form $T_i(\theta) = \theta + hV_i(\theta)$, where $V_i(\theta)$ is a random vector field on the parameter space depending on the randomly sampled batch of data $B_i$. (Of course, in the case of SGD we have $V_i(\theta) = -\nabla L_i(\theta)$ where $L_i = L_{B_i}$ is the loss function evaluated on batch $B_i$.)

- up to an error of order $\mathcal{O}(h^3)$ in the learning rate.

Unfortunately, it seems that in real-life cases higher orders beyond $\mathcal{O}(h^2)$ play a significant role in the backward dynamics, and therefore can not be neglected for reasonable ranges of the learning rate. Nevertheless, we include our second order approximation here, since we believe it gives a reasonable path on how to produce approximations of the backward trajectories using the forward trajectories. Namely, while this is out-of-scope for this paper and topic of further research, we believe that adding higher order corrections may produce useful approximations that could be used as stabilizers for the forward trajectories. The main idea is to expand a generic $k$ term iterate

$$T_{i_1} \cdots T_{i_k}(\theta) = (1 + hV_{i_1}) \cdots (1 + hV_{i_k})(\theta) \tag{6}$$

in Taylor's series in the learning rate. This is what the next lemma gives. (We demonstrate the usefulness of such expansions Appendix D where we use Lemma C.1 to produce a beneficial second order regularizer that emulates an iteration order average.)

Observe also that Lemma C.1 tells us that all possible iteration orders coincide at order $h$ but start differing at order $\mathcal{O}(h^2)$:

**Lemma C.1.** *Consider a sequence $\{T_i\}_{i>0}$ of operators of the form $T_i(\theta) = \theta + hV_i(\theta)$, where $V_i(\theta)$ is a vector field on the parameter space. Then for any choice of indices $i_1, \ldots, i_k$ we have that*

$$T_{i_1} \cdots T_{i_k} = 1 + h \sum_{l=1}^{k} V_{i_l} + h^2 \sum_{1 \le u < v \le k} V'_{i_u} V_{i_v} + \mathcal{O}(h^3)$$

*Proof.* We proceed by induction. For the base case, $k = 1$, this is trivial. Suppose now that this is true for any composition of $k-1$ operators. By definition of $T_{i_1}$ we have that

$$\begin{aligned} T_{i_1} \cdots T_{i_k}(\theta) &= T_{i_1}(T_{i_2} \cdots T_{i_k}(\theta)) \\ &= X + hV_{i_1}(X) \end{aligned} \tag{7}$$

with $X = T_{i_2} \cdots T_{i_k}(\theta)$. Now by induction hypothesis we have that

$$X = \theta + h \sum_{l=2}^{k} V_{i_l}(\theta) + h^2 \sum_{2 \le u < v \le k} V'_{i_u}(\theta)V_{i_v}(\theta) + \mathcal{O}(h^3) \tag{8}$$

Therefore, taking a Taylor series for the second term of (7), we obtain

$$hV_{i_1}(X) = hV_{i_1}(\theta) + h^2 \sum_{l=2}^{k} V'_{i_1}(\theta)V_{i_l}(\theta) + \mathcal{O}(h^3). \tag{9}$$

Summing up in (7) the expressions we have found for $X$ and $hV_{i_1}(X)$ above, we obtain that the composition $T_{i_1} \cdots T_{i_k}(\theta)$ has the form

$$\theta + h \sum_{l=1}^{k} V_{i_l}(\theta) + h^2 \sum_{1 \le u < v \le k} V'_{i_u}(\theta)V_{i_v}(\theta) + \mathcal{O}(h^3),$$

which completes the proof. □

The next theorem gives us a way to approximate the backward iterates up to order $\mathcal{O}(h^3)$ by correcting the forward iterates with a term keeping track of their difference:

**Theorem C.2.** *Consider a sequence $\{T_i\}_{i>0}$ of operators of the form $T_i(\theta) = \theta + hV_i(\theta)$, where $V_i(\theta)$ is a vector field on the parameter space. The backward and forward iterates of the sequence are related by the following identity:*

$$T_1 \cdots T_n(\theta) = T_n \cdots T_1(\theta) + h^2 \sum_{1 \le i < j \le n} [V_i, V_j](\theta) + \mathcal{O}(h^3)$$

*where the $[V_i, V_j](\theta) = V_i'(\theta)V_j(\theta) - V_j'(\theta)V_i(\theta)$ is the Lie bracket between the vector fields $V_i$ and $V_j$.*

*Proof.* By Lemma C.1, we have that the backward iterate is

$$T_1 \cdots T_n = 1 + h \sum_{l=1}^{n} V_l + h^2 \sum_{1 \le u < v \le n} V_u'V_v + \mathcal{O}(h^3)$$

while the forward iterate is obtained by reversing the indices:

$$T_n \cdots T_1 = 1 + h \sum_{l=1}^{n} V_l + h^2 \sum_{1 \le u < v \le n} V_v'V_u + \mathcal{O}(h^3).$$

We now see that the difference

$$D(\theta) = T_1 \cdots T_n(\theta) - T_n \cdots T_1(\theta)$$

between the backward and forward iterates is of the form

$$
\begin{aligned}
D(\theta) &= h^2 \sum_{1 \le u < v \le n} V_u'(\theta)V_v(\theta) - V_v'(\theta)V_u(\theta) + \mathcal{O}(h^3) \\
&= h^2 \sum_{1 \le u < v \le n} [V_u, V_v](\theta) + \mathcal{O}(h^3),
\end{aligned}
$$

which completes the proof. □

We now consider the case of SGD where the vector fields $V_i$ are given by the gradients of the loss evaluated at the current batch. We give below a definition of the approximate backward iterate in that particular case, although a similar definition can be given for any vector fields.

**Definition C.3.** Consider the SGD operators $T_i(\theta) = \theta - h\nabla L_i(\theta)$ obtained by taking the gradient of a loss function on a batch $B_i$ of data at step $i$. We denote by $\theta_n = T_n \cdots T_1(\theta_0) = \theta_{n-1} - h\nabla L_n(\theta_{n-1})$ the forward SGD iterate starting at initial point $\theta_0$ and by $\theta_n^B = T_1 \cdots T_n(\theta_0)$ the corresponding backward iterate starting at the same initial point. Motivated by Theorem C.2, we introduce the *approximate backward iterate $\tilde{\theta}_n$* as follows:

$$\tilde{\theta}_n = \theta_n + h^2 \sum_{1 \le i < j \le n} [\nabla L_i, \nabla L_j](\theta_0) \tag{10}$$

Theorem C.2 tells us that the true backward iterate and its approximation are within $\mathcal{O}(h^3)$ of each other:

$$\|\tilde{\theta}_n - \theta_n^B\| = \mathcal{O}(h^3), \tag{11}$$

which allows us to use the approximation for learning rates small enough so that the terms in $\mathcal{O}(h^3)$ can be neglected. The following corollary tells us how the approximate backward iterates can be computed in an iterative fashion by keeping in memory an additional variable $C_n$ of the same size as the network parameters:

**Corollary C.4.** *In the notation above, we can obtain the approximate backward SGD iterate $\tilde{\theta}_n$ from the forward SGD iterate $\theta_n$ starting at $\theta_0$ in an iterative fashion as follows:*

$$
\begin{aligned}
\theta_n &= \theta_{n-1} - h\nabla L_n(\theta_{n-1}) \\
g_n &= g_{n-1} + \nabla L_{n-1}(\theta_0) \\
H_n &= H_{n-1} + \nabla^2 L_{n-1}(\theta_0) \\
C_n &= C_{n-1} + H_n\nabla L_n(\theta_0) - \nabla^2 L_n(\theta_0)g_n \\
\tilde{\theta}_n &= \theta_n + h^2 C_n
\end{aligned}
$$

*with $g_0 = g_1 = 0$, $H_0 = H_1 = 0$, and $c_0 = c_1 = 0$.*

*Proof.* By Definition C.3 of the approximate backward iterates we have that $\tilde{\theta}_n = \theta_n + h^2 C_n$ with

$$
C_n = \sum_{1 \leq i < j \leq n} [\nabla L_i, \nabla L_j](\theta_0).
$$

First observe that we can split $C_n$ into two parts

$$
\begin{aligned}
C_n &= \sum_{1 \leq i < j \leq n-1} [\nabla L_i, \nabla L_j](\theta_0) + \sum_{1 \leq i \leq n-1} [\nabla L_i, \nabla L_n](\theta_0) \\
&= C_{n-1} + [\sum_{1 \leq i \leq n-1} \nabla L_i, \nabla L_n](\theta_0) \\
&= C_{n-1} + \left( \sum_{1 \leq i \leq n-1} \nabla^2 L_i(\theta_0) \right) \nabla L_n(\theta_0) \\
&\quad - \nabla^2 L_n(\theta_0) \left( \sum_{1 \leq i \leq n-1} \nabla L_i(\theta_0) \right) \\
&= C_{n-1} + H_n\nabla L_n(\theta_0) - \nabla^2 L_n(\theta_0)g_n,
\end{aligned}
$$

where $H_n$ and $g_n$ are expressed recursively as in the theorem statement. $\square$

## D  Approximate implicit regularization of smaller batches

As shown in Keskar et al. (2017); Smith et al. (2017; 2021); Dherin et al. (2022) for instance, small batches have an implicit regularization effect, producing solutions with higher test accuracy as the batch size decreases. In this section, we show that approximations of the type given by Lemma C.1 are useful to understand this implicit regularization effect. We also see how we can go beyond this small batch effect and produce explicit regularizers with even higher test performance based on the idea of iteration order average on the small batches, leveraging Lemma C.1.

Throughout the section we will write $V_i(\theta)$ for the negative batch loss gradient $-\nabla L_i(\theta)$ computed on batch $B_i$ in order to keep the notation simple.

### D.1  The effect of small batches

The idea is to use Lemma C.1 to understand the implicit regularization effect of small batches, very much in line with the findings in Smith et al. (2021). First, consider two settings. In the first setting, we perform a single gradient update

$$
T_{\text{large}}(\theta) = \theta + hV_B(\theta),
$$

with one large batch $B$ and learning rate $h$. In the second setting, we split the large batch $B$ into $c$ small batches of equal size: $B_1, B_2, \ldots, B_c$ (i.e, each of size $|B|/c$). Then we apply SGD sequentially $c$ times for

each of the smaller batch and with learning rate $h' = h/c$:

$$T_{\text{small}}(\theta) = (1 + h'V_c) \cdots (1 + h'V_2)(1 + h'V_1)(\theta)$$

Then Lemma C.1 gives us immediately a second order approximation for the second setting:

$$T_{\text{small}}(\theta) = \theta + h' \sum_{i=1}^{c} V_i(\theta) + h'^2 \sum_{i<j} V_i'(\theta)V_j(\theta) + \mathcal{O}(h^3)$$

Now it is easy to see that

$$hV_B(\theta) = h' \sum_{i=1}^{c} V_i(\theta).$$

Therefore, we can extract the added implicit regularization that is induced from smaller batches. Namely, we obtain that the composition of the $c$ smaller batches is the same as a single gradient step with the larger batch using learning rate $h = ch'$ plus an additional second-order regularization term:

$$T_{\text{small}}(\theta) = T_{\text{large}}(\theta) + \underbrace{h'^2 \sum_{i<j} V_i'(\theta)V_j(\theta)}_{\text{Smaller Batch Regularization}} + \mathcal{O}(h'^3)$$

So we can attribute the second order regularization effect of small batch training to this additional term, in line with the computations in Smith et al. (2021).

## D.2 Explicit regularization through iteration order averages

In the previous section, we applied the small batches in some chosen order. Another order may have worked as well, although producing a different second order regularization term. This begs the question of whether we can instead take the average over all possible iteration orders so as to produce an iterate for which no particular order is preferred:

$$T_{\text{perm}}(\theta) = \frac{1}{c!} \Big( \sum_{\sigma \in \text{Perm}_c} (1 + h'V_{\sigma(1)}) \cdots (1 + h'V_{\sigma(n)})(\theta) \Big), \tag{12}$$

where $\text{Perm}_c$ is the permutations of $c$ elements. Again Lemma C.1 gives us the second order approximation for this new update rule:

$$T_{\text{perm}}(\theta) = T_{\text{large}}(\theta) + \frac{1}{2}h'^2 \sum_{i \neq j} V_i'(\theta)V_j(\theta) + \mathcal{O}(h'^3).$$

In the next section, we show that the iteration-order-averaging term, that we can extract from the computation above, namely,

$$\lambda \sum_{i \neq j} V_i'(\theta)V_j(\theta),$$

produces a more powerful regularizer than the one obtained from a single ordering of the small batches. In a way, this new regularizer emulates a mixture of models, each of which is produced by a different ordering of the iterations in a window of $c$ batches (rather than from a different random seed). The next section shows the benefits of this new "ordering-free" regularizers experimentally.

### D.2.1 Experiments

We trained a MLP with 5 layers of 500 neurons each with stochastic gradient descent with no regularization to learn the 10 classes of Fashion-MNIST dataset Xiao et al. (2017). We used a learning rate of $h = 0.001$ (and $h' = h/c$), and a batch-size of $|B| = 2048^2$. We examine three different shrinking factors $c \in \{2, 3, 4\}$. In each setting, we test the values of $\lambda \in \{0.5h^2, h^2, 2h^2, 4h^2\}$. The following figures present the train and test loss over training compared to training with large-batch $|B| = 2048$ and learning rate $h' = h/c$, and training with small-batch $2048/c$ with learning rate $h'$. The experiment shows that our method can outperform vanilla training both for large and small batches with minimal tuning of $\lambda$ for this specific experiment.

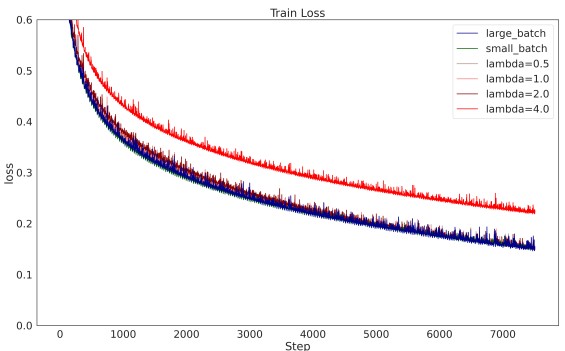

Figure 15: Train curves for injecting regularization resulting from splitting the batch in two (i.e., $c = 2$).

Figure 16: Test curves for injecting regularization resulting from splitting the batch in two (i.e., $c = 2$). As you can see, the performance improves as we increase the $\lambda$.

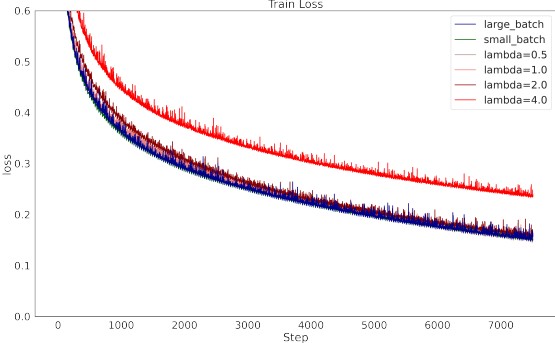

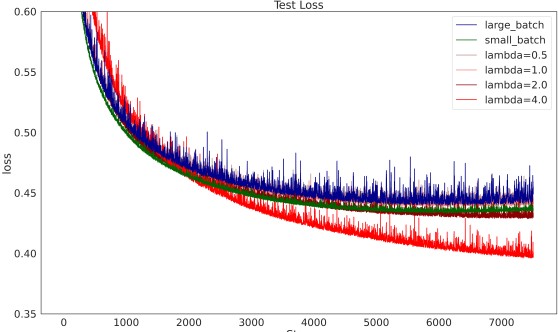

Figure 17: Train curves for injecting regularization resulting from splitting the batch in three (i.e., $c = 3$).

Figure 18: Test curves for injecting regularization resulting from splitting the batch in three (i.e., $c = 3$). Similar to the case where $c = 2$ above, the performance improves as we increase the $\lambda$.

---

[2]Or the closest integer divisible by $c$; e.g., 2049 for $c = 3$.

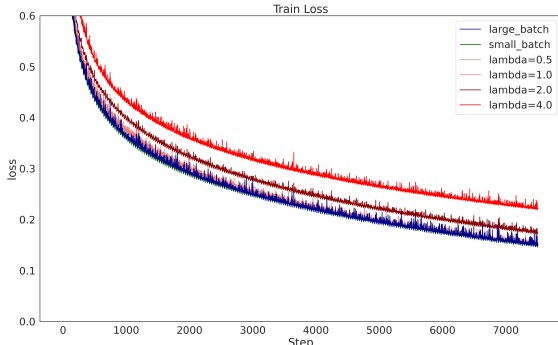 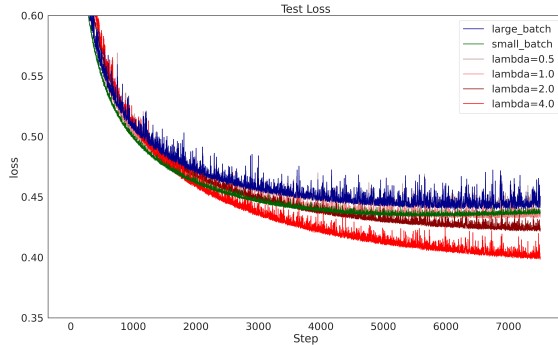

Figure 19: Train curves for injecting regularization resulting from splitting the batch in four (i.e., $c = 4$).

Figure 20: Test curves for injecting regularization resulting from splitting the batch in four (i.e., $c = 4$). Similar to the cases above, the trend continues and is even more visible; Performance improves as we increase the $\lambda$.

## E   Proof of Theorem 2.6

### E.1   Definitions

Before we dive into the proof, let us define the following three different notions of convergence of random variables. First we have the strongest notion, which is convergence almost surely. That is, a sequence of random variables $X_n$ converges to a limiting random variable $X_\infty$ almost surely if

$$\mathbb{P}(\lim_{n \to \infty} X_n = X_\infty) = 1.$$

This can be thought of as a sort of point-wise convergence. A weaker notion of convergence which is implied by almost sure convergence (see Theorem 2.7 (i) in Van der Vaart (2000)) is convergence in probability. The sequence $X_n$ of random variables converges to the random variable $X_\infty$ in probability if for every $\varepsilon > 0$ the following holds

$$\lim_{n \to \infty} \mathbb{P}(|X_n - X_\infty| > \epsilon) = 0.$$

That is, for large $n$ it becomes more and more likely that $X_n$ is close to $X_\infty$. The weakest notion of convergence is convergence in distribution, which is implied by convergence in probability (see Theorem 2.7 (ii) in Van der Vaart (2000)). There are many equivalent ways to define this, but the most concrete way is the following: Let $F_n(x) = \mathbb{P}(X_n \leq x)$ be the cumulative distribution function for $X_n$ and let $F_\infty(x)$ be the corresponding cumulative distribution function for $X_\infty$, then we say that the sequence $X_n$ converges in distribution to $X_\infty$ if for every continuity point $x$ of the cumulative distribution function $F_\infty$ we have the following

$$\lim_{n \to \infty} F_n(x) = F_\infty(x).$$

If we define $\mu_n := dF_n$ and $\mu_\infty := dF_\infty$, then when we say that $\mu_n \to \mu_\infty$ we mean convergence in distribution in the above sense.

### E.2   Proof

Consider a sequence $T_i : \Omega \to \Omega$, $i = 1, 2, \ldots$ of independent and identically distributed random operators. The probability distribution of the forward iterates $\theta_n = T_n T_{n-1} \cdots T_1(\theta_0)$ is given by

$$\mu_n = (T_n T_{n-1} \cdots T_1)_* \delta_{\theta_0},$$

where $\delta_{\theta_0}$ is the delta distribution concentrated at $\theta_0$ (i.e. $\delta_{\theta_0}(A) = 1$ if $\theta_0 \in A$ and 0 otherwise).

We want to show that when the backward iterates converge to a random point (randomness is due to the sampling of the random operators $T_i$'s), i.e., when

$$T_1 T_2 \cdots T_n(\theta_0) \longrightarrow \theta^* \quad \text{as} \quad n \to \infty,$$

then this implies that

- the probability distribution of the forward iterates from $\theta_0$ converge to a stationary probability measure $\mu_n \to \mu_{\theta^*}$

- the random point $\theta^*$ is distributed according to the same forward iterate stationary distribution $\mu_{\theta^*}$.

To see that let $Y_n^{\theta_0}$ denote the forward iterates

$$Y_n^{\theta_0} = T_n T_{n-1} \ldots T_1(\theta_0),$$

and let $X_n^{\theta_0}$ denote the backward iterates

$$X_n^{\theta_0} = T_1 T_2 \ldots T_n(\theta_0).$$

Define the cumulative distribution function (CDF) for the random vector $Y_n^{\theta_0}$ and $\theta = (\theta^1, \ldots, \theta^d) \in \mathbb{R}^d$ as

$$F_n^{\theta_0}(\theta) := \mathbb{P}((Y_n^{\theta_0})_1 \leq \theta^1, \ldots, (Y_n^{\theta_0})_d \leq \theta^d).$$

By independence of the random operators $T_i$'s we know by symmetry that $F_n^{\theta_0}(\theta)$ is also the CDF of $X_n^{\theta_0}$.

Under our assumption that the backward iterates converge to a random point $\theta^*$ (almost sure convergence) we will define the CDF of that random point as, $F_\infty^{\theta_0}(\theta)$ and for the sake of the proof we denote that point with $X_\infty^{\theta_0}$ instead of $\theta^*$ to highlight that it is a random vector.

Now, our assumption that the backward iterates converge to a point implies that the random vector $X_n^{\theta_0}$ converges to $X_\infty^{\theta_0}$ almost surely, in the sense defined above. As alluded to in the previous section above, the almost sure convergence implies convergence in distribution, i.e., the convergence of the CDF $\lim_{n \to \infty} F_n^{\theta_0}(\theta) = F_\infty^{\theta_0}(\theta)$, for every $\theta$ where $F_\infty^{\theta_0}$ is continuous.

Since $X_n^{\theta_0}$ and $Y_n^{\theta_0}$ have the same distribution (same CDF) we have immediately that $Y_n^{\theta_0}$ also converges in distribution to $X_\infty^{\theta_0}$. The limiting stationary distribution $\mu_{\theta^*}$ is simply the probability measure corresponding to $F_\infty^{\theta^*}$, i.e. $\mu_{\theta^*} := dF_\infty^{\theta_0}$.

## F   Proof of Lemma 2.5

Consider a smooth function $L : \mathbb{R}^d \to \mathbb{R}$ satisfying (3) and (4).

We want to show the following inequality

$$\|T(\theta_1) - T(\theta_2)\| \leq \sqrt{1 - 2hm + h^2 M^2} \, \|\theta_1 - \theta_2\| \, .$$

for the gradient operator $T(\theta) = \theta - h\nabla L(\theta)$.

We can see that by first expanding the square of the operator difference:

$$\begin{aligned}
\|T(\theta_1) - T(\theta_2)\|^2 = \|\theta_1 - \theta_2\|^2 \\
- 2h\langle \nabla L(\theta_1) - \nabla L(\theta_2), \theta_1 - \theta_2 \rangle \\
+ h^2 \|\nabla L(\theta_1) - \nabla L(\theta_2)\|^2.
\end{aligned}$$

Now applying the strict convexity conditions, we get

$$\begin{aligned}
\|T(\theta_1) - T(\theta_2)\|^2 &\leq \|\theta_1 - \theta_2\|^2 - 2hm\|\theta_1 - \theta_2\|^2 \\
&+ h^2 M^2 \|\theta_1 - \theta_2\|^2 \\
&= (1 - 2hm + h^2 M^2)\|\theta_1 - \theta_2\|^2,
\end{aligned}$$

which ends the proof by taking the square root on both sides.

# G   Discussion on the assumption of contractions.

The following discussion demonstrates that the contraction hypothesis required in Theorem 2.2 can be satisfied under standard conditions that frequently arise in deep learning. Specifically, we show that gradient descent transformations are contractions with respect to a natural loss-based pseudo-metric when the loss function satisfies both Lipschitz and Polyak-Łojasiewicz conditions.

**Setup and Assumptions**   Let $S$ be a domain in $\mathbb{R}^M$ representing the parameter space, and let $L : S \to \mathbb{R}_{\geq 0}$ be a *loss function* to be minimized, having the form

$$L(\theta) = \frac{1}{m} \sum_{i=1}^{m} l_i(\theta).$$

We make the following standard assumptions:

1. Each individual loss $l_i \geq 0$ and the global minimum value of $L$ is 0. This *interpolation* assumption is reasonable for over-parametrized systems where the model can perfectly fit the training data.

2. All functions are differentiable.

3. There exists a constant $C$ such that all gradients $\nabla l_i$ (and therefore $\nabla L$) are $C$-Lipschitz continuous in the relevant parameter space $S$.

**The Polyak-Łojasiewicz Condition**   Additionally, we assume that $L$ satisfies the *Polyak-Łojasiewicz (PL)* condition: there exists a constant $\mu > 0$ such that

$$\frac{1}{2} \left\| \nabla L(\theta) \right\|^2 \geq \mu L(\theta)$$

for all $\theta \in S$. This condition, while satisfied by strongly convex functions, is particularly important because it also holds for over-parametrized neural networks (which are typically non-convex), as demonstrated in Liu et al. (2022). The PL condition essentially ensures that the gradient magnitude provides a lower bound on the distance to optimality.

**Loss Pseudo-Metric**   To analyze the contraction properties of gradient-based updates, we introduce a natural pseudo-metric based on the loss function.

**Definition.** For the parameter space $S$, we define the *loss pseudo-metric* $d : S \times S \to \mathbb{R}_{\geq 0}$ by:

$$d(x, y) = \begin{cases} L(x) + L(y) & \text{if } x \neq y, \\ 0 & \text{if } x = y. \end{cases}$$

**Properties.** This function satisfies the standard metric axioms:

1. **Symmetry:** $d(x, y) = L(x) + L(y) = L(y) + L(x) = d(y, x)$.

2. **Triangle inequality:** For any $x, y, z \in S$:

$$d(x, y) = L(x) + L(y) \leq L(x) + L(z) + L(z) + L(y) = d(x, z) + d(z, y),$$

which follows from the non-negativity of $L$.

3. **Pseudo-metric property:** $d(x, y) = 0$ for $x \neq y$ if and only if $x, y \in Z$, where $Z := \{\theta \in S : L(\theta) = 0\}$ denotes the set of global minima.

The key insight is that $d$ measures the "total suboptimality" of two points. Unlike a true metric, distinct points in the zero-loss set $Z$ have distance zero from each other, hence the term *pseudo*-metric.

**Contraction Analysis for Full-Batch Gradient Descent**  We now establish that gradient descent is a contraction in the loss pseudo-metric $d$. The Lipschitz condition on $\nabla L$ immediately implies that for any $x, y \in S$:

$$L(y) - L(x) = \int_0^1 \nabla L(x + t(y - x)) \cdot (y - x) \, dt \leq \langle \nabla L(x), y - x \rangle + \frac{C}{2} \|y - x\|^2.$$

Consider the full-batch gradient descent transformation $T(x) = x - h\nabla L(x)$ with learning rate $h > 0$. Applying the above inequality with $y = T(x)$, we obtain:

$$
\begin{aligned}
L(T(x)) - L(x) &\leq \langle \nabla L(x), T(x) - x \rangle + \frac{C}{2} \|T(x) - x\|^2 \\
&= \langle \nabla L(x), -h\nabla L(x) \rangle + \frac{C}{2} \|-h\nabla L(x)\|^2 \\
&= -h\left(1 - \frac{Ch}{2}\right) \|\nabla L(x)\|^2.
\end{aligned}
$$

Combining this with the PL condition and requiring $0 < h < 2/C$ (a standard step-size constraint), we get

$$L(T(x)) - L(x) \leq -h\left(1 - \frac{Ch}{2}\right) \|\nabla L(x)\|^2 \leq -2h\mu\left(1 - \frac{Ch}{2}\right) L(x).$$

This yields the key contraction result:

$$L(T(x)) \leq (1 - h\mu(2 - Ch)) L(x)$$

for sufficiently small $h$. The factor $k := 1 - h\mu(2 - Ch) < 1$ represents the contraction rate. That is, for two points $x, y \in S$, we have

$$d(Tx, Ty) = L(Tx) + L(Ty) \leq kL(x) + kL(y) = kd(x, y),$$

establishing that $T$ is a contraction in the loss pseudo-metric $d$.

**Extension to Stochastic Mini-Batch Updates**  For stochastic gradient descent with mini-batches, we consider transformations of the form

$$T_i(x) = x - h\nabla\left(\frac{1}{m} \sum_{j \in I_i} l_j\right)(x),$$

where $I_i$ denotes a mini-batch (a subset of indices) of size $m$. Through a similar to above but more technical argument (see (Bassily et al., 2018, p. 3-4)), one can show that

$$\mathbb{E}\left[L(T_i(x))\right] \leq kL(x),$$

for some contraction factor $0 < k < 1$, provided the hyperparameters are chosen appropriately. The key insight is that while individual mini-batch updates may not be contractions, they are contractions *in expectation*.

**Application to the Backward Contraction Principle**  The contraction analysis established above directly enables the application of our backward contraction principle (Theorem 2.2) to gradient-based optimization. Several technical points ensure the theory applies seamlessly:

- **Pseudo-metric compatibility:** The contraction principle extends naturally from true metrics to pseudo-metrics. In our loss pseudo-metric framework, the zero-loss set $Z$ effectively becomes a single point when passing to the metric completion, serving as the unique fixed point of both $T$ and any $T_i$.

- **Bounded displacement condition:** The required condition (2) can be satisfied by restricting the parameter space $S$. For instance, imposing $L(\theta) \leq B$ for some bound $B$ creates an invariant set since gradient descent decreases $L$ (and $T_i$ decreases $\mathbb{E}[L]$).

- **Stochastic contractions:** For mini-batch updates that are only contractions in expectation, more sophisticated analysis techniques are available (see Diaconis & Freedman (1999) and references therein).

This framework provides the rigorous foundation needed to apply our backward iteration theory to practical deep learning optimization algorithms.

