# OpenReview forum: "How iteration composition influences convergence and stability in deep learning"
_TMLR — Accepted by TMLR_

### Review · Reviewer_Pyai · 2025-08-25

**Summary Of Contributions:**

**Summary**

This paper studies how reversing the order of samples in stochastic gradient methods affects the algorithm's convergence behavior. The authors show that their backward iteration has better stability properties compared to the standard SGD update by analyzing the contraction of the associated random operators. Some numerical experiments validate the effect of the proposed method.

**Audience:**

Yes

**Audience Explanation:**

**Strength**

This paper is, in general, easy to follow. The perspective of running backward iteration looks interesting, and some experiments show its effect. I believe this paper would be of interest to the community if my concerns are addressed.

**Broader Impact Concerns:**

N.A.

**Claims And Evidence:**

No

**Claims Explanation:**

**Weaknesses**

Although the perspective is interesting, I have several concerns regarding its relevance to real neural network training, and more broadly, stochastic optimization.

1. Theoretical result

   One of the main results of the paper is the claim that "backward update" has better stability properties compared to forward update. The idea of the paper can be summarized as follows: since the backward update injects randomness to the beginning instead of the end of the trajectory, the algorithm will finally be dominated by the initial samples, say $\{B_1, B_2, \ldots, B_{M}\}$. Given the assumptions in this paper on the random operators, the "convergence of iterates" follows almost immediately.

   However, I have reservations about its significance/soundness.

   First, I feel it's unreasonable to treat each "backward iteration" as an algorithm update, since generating "backward iteration $k$" involves running SGD through the entire algorithm trajectory over $k$ samples; computations from all the previous iterations are discarded. Therefore, the paper essentially studies the relation between the last iterates of different SGD trajectories, given that the finally encountered samples -- $\{B_1, B_2, \ldots, B_{M}\}$ are the same. In my opinion, this approach is not directly comparable to SGD.

   Besides, I also find that the claimed "stability" result unconvincing. For example,

   - The paper uses operator notation throughout and applies a *local* second-order Taylor expansion $T(\theta) = \theta^\star + (1 - h \nabla^2 L (\theta^\star))(\theta - \theta^\star)$ in several places. In other words, several results are discussed almost in convex quadratic case. However, real neural networks are typically not smooth or convex.
   - Even under twice differentiablity assumption, to ensure that the local expansion (uniform contraction) holds, the algorithm has to stay in a small neighbourhood around $\theta^\star$. In the context of SGD, it imposes several additional restrictions: the action of previous operators need to ensure that the iterates stay locally close to $\theta^\star$ so that uniform contraction still holds in later iterations. It contrasts the claimed stability argument.

   Finally, from an optimization perspective, the cost of *the last* "backward iterate" is exactly the same as SGD with reversed sampling order. When the stochastic noise is i.i.d., the backward iteration would give  no gain in terms of convergence rate. This can be seen from the quadratic loss **Example 3.1**.

2. Practicality of the result and numerical experiments

   The authors mentioned in the paper that a backward iteration is very expensive. As mentioned in the previous point, this is because each backward iteration is simply running SGD from scratch and generating a new trajectory. Therefore, given the independence between backward iterates, the experiment seems to suggest that "if we run SGD with reversed sampling order, then its performance is better", which is counterintuitive to me.

Besides, I also find the structure of the paper slightly confusing. For example, there are several examples in the paper (**Section 2.1, 3.1, 3.2**), but they are either quadratic or a local approximation to quadratic. There are unnecessary high-level discussions on the extensions, such as **Remark 2.6**.

Given the restrictions above, I believe this paper should be largely rewritten to improve its presentation and to clarify its contribution.

**Requested Changes:**

**Questions**

1. Could you clarify why each backward iteration should be viewed as an iteration, instead of the last iterate of a new optimization trajectory?
2. Can the theory be extended beyond the smooth convex case? How could you ensure that the local assumptions in the paper are satisfied by the whole iteration trajectory when there is noise? How to choose $h$ in practice?
3. The current description of backward iteration is highly inefficient since most of the information from previous iterations is discarded. Could you find a way to reuse them?
4. Could you add an experiment where the initial samples ($\{B_1, B_2, \ldots, B_{M}\}$) have higher variance than later samples and see whether the backward method still performs well?

**Minor issues**

1. Page 2

   "aiming to reduce the training loss on batch $B_i$" is not accurate

2. Page 2

   $\theta$ in the randomly ...  => $\theta$ is the randomly ...

3. Page 2

   Computation intensive => Computationally intensive

4. Page 4

   Please use $I$ instead of $1$ for identity matrix

5. Page 6

   strictly convex => strongly convex

6. Page 6

   Remark 2.6 looks irrelevant to the topic of the paper

7. Page 8

   a i.i.d. => an i.i.d.

8. Page 9

   $\lambda_{max}$ =>  $\lambda_\max$

9. Page 10, 11,

   Please align the figures in a row

**Requested changes**

See questions and minor issues

---

> ### Author Response · Authors · 2025-09-04
>
> Thank you for your time and for providing a detailed and insightful review of our paper. We are grateful for your positive feedback that the paper is "well-written and easy to follow" and that you find our perspective "interesting". We have carefully considered your concerns and here is our best effort to address them.
>
> Before diving into your specific questions though, let us remind that our central contribution is to introduce and analyze a fundamental, and to our knowledge, previously unexplored aspect of stochastic optimization: the influence of the update composition order on the training trajectory's stability and convergence properties. We show theoretically that the "backward trajectory," where iterates are produced by composing updates in reverse, converges to a point, whereas the standard "forward trajectory" converges to a distribution. Our main goal is **not** to propose backward SGD as a new, practical, drop-in replacement for standard optimizers, but rather to use it as a tool to reveal this underlying phenomenon and open up new research avenues for improving training stability.

---

> ### Author Response · Authors · 2025-09-04
> **Question 1**
>
> > *Question 1: Could you clarify why each backward iteration should be viewed as an iteration, instead of the last iterate of a new optimization trajectory?*
>
> This is an excellent question that strikes at the core of our paper's perspective.
>
> When we refer to the "$n$-th backward iterate," we are referring to the specific parameter vector, $\theta_n^B = T_1 T_2 \cdots T_n(\theta_{init})$, produced at step $n$. Our work is focused on analyzing the mathematical properties (e.g., stability, convergence) of the *entire sequence* of these vectors, i.e., the backward trajectory $\{\theta_1^B, \theta_2^B, \dots, \theta_N^B\}$, and comparing it to the sequence of forward iterates, $\{\theta_1^F, \theta_2^F, \dots, \theta_N^F\}$.
>
> The main contribution of this paper is to prove that to any forward trajectory obtained by a process of operator composition in the forward way, a more stable trajectory exists: the backward trajectory. We mathematically construct the backward trajectory by reversing the composition order of the forward iterates, and use this mathematical construction of the backward trajectory in our proof. While this particular construction of the backward trajectory is not meant as a new competitive algorithm, we show a clear path to practicality in Appendix C where we derive a recursive approximation. This method computes the stable backward iterate by applying a small, cheap correction term to the standard forward iterate, allowing for re-use of the previous computation, and demonstrating how to harness the stability benefits without the prohibitive cost.
>
> This duality between backward and forward trajectories has a long history in mathematics, which seems to be unknown to optimization theory. For instance, the process of continued fraction expansions on how to best approximate a real number by a rational number is another example of a naturally a backward iteration  (see https://en.wikipedia.org/wiki/Continued_fraction under “The continued fractions as a composition of linear fractional maps”) .

---

> > ### Comment · Reviewer_Pyai · 2025-10-01
> > **Thank you for the response**
> >
> > Thank you for the response. The authors address some of my concerns, but I still find the argument a bit unconvincing: as I initially commented, when the samples are fed into the algorithm in the backward order and under the assumption of uniform contraction, it is clear that the trajectory will be dominated by the initial samples (which are the last samples when generating the trajectory). The reason is simply that under uniform contraction, initialization would have a vanishing impact on the algorithm trajectory. While this argument seems interesting, its application still looks quite restrictive to me.
> >
> > Given the remaining concerns, I currently recommend borderline rejection. But I won't object if other reviewers decide to accept the paper.
> >
> > **Unaddressed issues**
> >
> > 1. The definition of strict convexity is still not correct [1].
> > 2. Please align the figures in a row. The current layout looks strange.
> >
> > [1] Nesterov, Y. (2013). *Introductory lectures on convex optimization: A basic course* (Vol. 87). Springer Science & Business Media.

---

> > > ### Author Response · Authors · 2025-10-02
> > >
> > > Thank you for your prompt response. We are of course very happy to fix the remaining two unaddressed issues.

---

> ### Author Response · Authors · 2025-09-04
> **Question 2**
>
> > *Question 2: Can the theory be extended beyond the smooth convex case? How could you ensure that the local assumptions in the paper are satisfied by the whole iteration trajectory when there is noise? How to choose $h$ in practice?*
>
>
> These are important questions regarding the scope of our theory.
>
>
> **Extending Beyond the Convex Case:**
> Our formal proof of convergence relies on the uniform contraction principle (Theorem 2.2), which is most easily established in smooth, strongly convex landscapes. We acknowledge that deep learning loss landscapes are far more complex. However, our empirical results across multiple non-convex architectures and datasets consistently show that backward trajectories are more stable throughout training—not just near minima. This strong empirical validation suggests a more general principle is at work. In fact, it is possible to extend our Theorem 2.2 to the non-convex setting, but at a heavy theoretical cost. We now revised our paper to explain this: We removed Remark 2.6 as you suggested, and instead added Appendix G that explains in details how to extend our theorem to the deep learning setting. Remark 2.3 has also been updated to account for the presence of the new Appendix G. The gist of the argument is as follows:
>
>
> * If we are willing to introduce “pseudo-metrics” rather than metrics and use a contraction property “on average” for these pseudo metrics, we can generalize our theorem in the case where the loss function $L(x)$ satisfies a Polyak-Lojasiewicz condition, together with a Lipschitz condition on the gradients ([1] argues that this is the case in deep-learning in large regions of the loss landscape). In the interpolation case, this means the gradient updates are contractions on average in the pseudo-metric d(x,y):=L(x)+L(y). (Pseudo refers to that the distance between interpolation minima is 0 even if these minima do not coincide). Our backward contraction mapping principle therefore basically applies to neural network settings beyond the convex case.
>
> **Ensuring Local Assumptions Hold:** You are correct that noise and the global nature of training make it difficult to guarantee that iterates remain in a region where local assumptions hold. Our current theory does not cover this. The surprising empirical result, however, is that the stability benefit *does* manifest globally. We see this as a key finding of our paper: the backward composition provides a stabilizing effect that is more powerful than our current, localized theory can explain. Note that our generalized argument in the new Appendix G assuming the PL condition implies a contractivity property of the batch gradients only *on average*, which is enough.
>
> **Choosing $h$:** In our work, the learning rate $h$ is chosen in the same way as in any standard training procedure. Our theory does not prescribe the size of the learning rate, even in the backward trajectory case. Having a theory prescribing useful learning rate ranges in deep-learning is a hard open problem at the time of writing, beyond the scope of this paper.
>
> [1] Liu, Chaoyue; Zhu, Libin; Belkin, Mikhail, Loss landscapes and optimization in over- parameterized non-linear systems and neural networks, Applied and Computational Harmonic Analysis. Special Issue on Harmonic Analysis and Machine Learning. 59 (2022) 85-116

---

> ### Author Response · Authors · 2025-09-04
> **Question 3**
>
> > *Question 3: The current description of backward iteration is highly inefficient since most of the information from previous iterations is discarded. Could you find a way to reuse them?*
>
> This is a crucial point regarding practical viability. The naive implementation we describe is indeed intended as a conceptual tool to study the backward trajectory, not as a practical algorithm.
>
> However, a path to a more efficient, recursive implementation is possible. In **Appendix C**, we derive an approximation (at second order in $h$) that computes the backward iterate by adding a correction term to the standard (and cheap) forward iterate. Specifically, Theorem C.2 and Corollary C.4 provide an explicit, iterative formula for a second-order approximation:
>
> $\theta_n^B \simeq \theta_n^F+h^2\sum_{1\le i<j\le n}[\nabla L_i,\nabla L_j]$
>
> This shows a clear path to reusing past computations (via the forward iterate $\theta_n^F$) and developing practical backward-inspired algorithms. While we note that this second-order approximation may not be sufficient in practice, it establishes a valid research direction for making these stable trajectories computationally accessible. We will elevate this discussion to the main paper to make this point more clearly.

---

> ### Author Response · Authors · 2025-09-04
> **Question 4**
>
> > *Question 4: Could you add an experiment where the initial samples have higher variance than later samples and see whether the backward method still performs well?*
>
> A key assumption of our theory which is an essential ingredient of our proof is that the stochastic operators need to be independently and equally distributed. This is no longer the case if the initial samples have a larger variance than the rest of the batches, and our theory does not apply to this case.

---

### Review · Reviewer_rt2N · 2025-09-12

**Summary Of Contributions:**

The paper studies how the *order* of composing stochastic update maps affects optimization. It defines a “backward” trajectory that applies the per-batch update maps in reverse order, and proves a *backward contraction principle* showing pointwise convergence of the backward iterates under uniform contraction assumptions (Theorem 2.2). It also relates the backward limit to the stationary distribution of the forward Markov chain (Theorem 2.7). Simple models illustrate why noise decays in the backward order, and experiments on CIFAR-10/100 and Fashion-MNIST with small batches and fixed learning rates suggest improved stability of the backward trajectory. The paper discusses the prohibitive cost of the naive backward scheme and proposes an analytical approximation using Lie-bracket corrections (Appendix C).

**Audience:**

Yes

**Audience Explanation:**

Iteration order and stability are timely topics for the optimization and generalization communities. The paper offers a crisp theoretical lens (backward vs. forward compositions) and empirical patterns (reduced variance across seeds on CIFAR/Fashion-MNIST under small-batch, constant-LR SGD) that many readers will find thought-provoking even if not yet practice-ready.

**Broader Impact Concerns:**

The submission states it does **not** foresee negative impacts.

**Claims And Evidence:**

No

**Claims Explanation:**

The paper’s core theoretical claims (e.g., the Backward Contraction Mappings Principle, Thm. 2.2) are stated clearly and proved correctly under their assumptions, and the link between backward limits and the forward stationary distribution (Thm. 2.7) is sound.

However, the assumptions (uniform contraction; i.i.d. random operators; bounded displacement) are strong and not validated in realistic deep-net settings, and the empirical section, while suggestive, does not control for the large extra compute of naive backward recomposition, which confounds claims of “faster” or “more stable” convergence relative to forward SGD. Finally, the Lie-bracket approximation is presented theoretically but its practicality is not demonstrated. Overall, the evidence is insightful but not yet convincing for the broad claims about stability/convergence advantages in practice.

**Requested Changes:**

1. **Compute-normalized experimental comparisons.**
   Compare backward and forward under equal FLOPs/wall-clock (or report TFLOPs/energy) to separate algorithmic stability from extra compute due to recomposition. Include windowed/backward-late variants at matched budgets;

2. **Quantitative stability metrics and statistics.**
   Beyond per-seed plots, report per-seed variance of loss/accuracy, oscillation amplitude, integrated path length, and confidence intervals. Summarize test accuracy distributions and run basic statistical tests across seeds;

3. **Baseline controls that use similar extra compute.**
   Add cheap, compute-matched baselines: (i) tail-replay of last *M* batches; (ii) EMA/SWA; (iii) gradient accumulation / multiple passes on the same mini-batch; (iv) intermittent short-window reversal;

4. **Clarify scope/assumptions and provide diagnostics.**
   Discuss when contraction-like conditions plausibly hold in deep nets; include empirical diagnostics (e.g., local Lipschitz/contractivity estimates around late-training iterates) and comment on i.i.d. vs. epoch-shuffle dependence.;

5. **Practicality of the Lie-bracket approximation.**
   Provide complexity/memory analysis and an implementation using Hessian-vector products; show accuracy vs. cost and sensitivity to learning rate (since higher-order terms appear to matter). Include ablations of order-2 vs. higher-order corrections, or clearly delimit why higher order is needed.

6. **Coverage of optimizers and settings.**
   Add results with momentum/AdamW and at least one non-vision model (e.g., a small transformer) to test generality; report behavior with LR schedules and standard augmentations.

7. **Reproducibility and clarity.**
   Release code and add concise pseudocode for: (a) naive backward; (b) windowed/reset/backward-late; (c) approximate backward update (caching & Hessian-vec usage). Specify data-loader policy (with/without replacement), seeds, and evaluation cadence.


8. **Positioning vs. related ideas.**
   Briefly connect the approximation to BCH-type expansions; relate the empirical stability narrative to “edge of stability” and curriculum/order effects discussed in the intro.

9. **Minor edits.**
   Clean up typos and unify notation (e.g., identity element formatting) and figure captions. (Examples occur around App. C and the figure appendix.)

---

> ### Author Response · Authors · 2025-09-26
>
> Thank you for your insightful review. We are very pleased that you found our theoretical results "clearly stated and proved correctly" and that "the paper offers a crisp theoretical lens [...] that many reader will find thought-provoking”.
>
>
> The concern regarding the *restrictiveness of the uniform contraction assumption* is well-taken. In the standard Euclidean metric, this assumption is indeed too strong for non-convex deep learning settings. To address this gap, we have:
>
> 1. *Added a new discussion in Appendix G*, and
> 1. *Updated Remark 2.3* to reflect this broader perspective.
>
> In Appendix G, we show that our *backward contraction principle (Theorem 2.2)* holds in a more general and relevant setting for deep learning. This requires introducing the abstract notion of *pseudo-metric spaces*, which allows us to extend the contraction argument beyond convexity.
>
> Specifically, we demonstrate that if the loss satisfies the *Polyak–Łojasiewicz (PL) condition*, which, as argued by Liu et al. (2022), is often satisfied in over-parameterized neural networks. In this setting, the gradient updates are *contractions on average* with respect to a *loss pseudo-metric* defined as $d(x,y)=L(x)+L(y)$. We then show that our backward contraction argument carries out in this generalized setting as well.
>
>
> This extension significantly broadens the applicability of our theoretical results, providing a formal explanation for the stabilizing effects observed in our experiments, even in non-convex deep learning scenarios.
>
>
> Regarding the *computational cost of naive backward-SGD*, we fully agree: it is prohibitive in practice. We want to reiterate that we are *not proposing backward SGD as a competitive algorithm.* We are using it to access the backward trajectory in order to demonstrate theoretically its increased stability. Our goal is to establish a *theoretical foundation* for the iteration composition approach. Even the approximation presented in *Appendix C* remains expensive when using exact Hessians.  The approximation in Appendix C is intended as a *proof of concept*, not a deployable method.
>
>
> While techniques such as *vector products* or *diagonal approximations* could make the method more tractable, we do not pursue these directions in this paper. Developing practical, low-cost algorithm that leverages these stability properties remains an open research problem beyond the scope of this paper.
>
> We hope these clarifications help contextualize our contributions and delineate the theoretical nature of our work from practical optimization proposals.

---

> > ### Author Response · Authors · 2025-09-27
> > **Questions**
> >
> > >*Question 1: Compare backward and forward under equal FLOPs [...] to separate algorithmic stability from extra compute [...]*
> >
> > We agree that naive backward-SGD is computationally prohibitive, and even the approximation in Appendix C remains costly when using exact Hessians. While compute-normalized comparisons could be informative, we consider them out of scope for this paper, as our goal is not to propose backward-SGD as a practical optimizer. Instead, we use it as a *conceptual tool* to reveal the impact of iterate composition on stability.
> >
> > >*Question 2: Beyond per-seed plots, report per-seed variance of loss/accuracy, oscillation amplitude, integrated path length, [...].*
> >
> > We believe that these additions would shift the focus away from the theoretical contributions.
> >
> > >*Question 3: Add cheap, compute-matched baselines: (i) tail-replay of last M batches; (ii) EMA/SWA; (iii) gradient accumulation / multiple passes on the same mini-batch; (iv) intermittent short-window reversal.*
> >
> > We appreciate the suggestion. However, since we are *not proposing backward-SGD as a practical alternative*, compute-matched baselines are not central to our scope. Our aim is to isolate and understand the effects of iterate composition order, not to benchmark against practical stabilizers. We will clarify this in the manuscript.
> >
> > >*Question 4: Discuss when contraction-like conditions plausibly hold in deep nets; include empirical diagnostics (e.g., local Lipschitz/contractivity estimates around late-training iterates) and comment on i.i.d. vs. epoch-shuffle dependence.*
> >
> > This is a crucial point. The uniform contraction assumption is indeed too strong in the Euclidean metric for deep learning. To address this, we have added a new discussion in *Appendix G* and updated *Remark 2.3*. We show that our backward contraction principle (Theorem 2.2) holds under the *Polyak–Łojasiewicz (PL) condition*, which is often satisfied in over-parameterized networks. This leads to *average contraction* in a pseudo-metric space defined by the loss. While this introduces technical complexity, it significantly broadens the applicability of our theory. We will also clarify the role of the i.i.d. assumption, which remains standard and reasonable in our setting.
> >
> > >*Question 5: Provide complexity/memory analysis and an implementation using Hessian-vector products; show accuracy vs. cost and sensitivity to learning rate. Include ablations of order-2 vs. higher-order corrections, or clearly delimit why higher order is needed.*
> >
> >
> > We agree that the approximation in Appendix C is costly and not yet practical. We have clarified that this is a *proof of concept*, intended to demonstrate that such approximations are possible. We are not claiming that the method is ready for deployment. While Hessian-vector products and diagonal approximations could reduce cost, we leave this for future work. We will revise the manuscript to emphasize that developing a practical, low-cost algorithm is an *open research problem*, and that our current focus is on establishing a possible path forward.
> >
> >
> > >*Question 6: Add results with momentum/AdamW and at least one non-vision model (e.g., a small transformer) to test generality; report behavior with LR schedules and standard augmentations.*
> >
> >
> > We appreciate the suggestion. However, broader coverage of optimizers and architectures is *outside the scope* of this paper. Our focus is on the theoretical implications of iterate composition order, and the experiments are designed to support that narrative, and not propose a new algorithm competitive with standard approach. We will clarify this in the revised manuscript.
> >
> >
> > >*Question 7: Release code and add concise pseudocode for: (a) naive backward; (b) windowed/reset/backward-late; (c) approximate backward update (caching & Hessian-vec usage). Specify data-loader policy (with/without replacement), seeds, and evaluation cadence.*
> >
> > We agree that reproducibility is important. We will add *concise pseudocode* for the key variants.
> >
> > >*Question 8: Briefly connect the approximation to BCH-type expansions; relate the empirical stability narrative to “edge of stability” and curriculum/order effects discussed in the intro.*
> >
> > There is a connection between the BCH formula and our approximation. However we fail to see its significance in this context. We are curious to understand what you have in mind. The connection is as follows: The forward iterates can be considered as an approximation of
> > $$\frac 1hBCH(hV_n,\frac 1hBCH(hV_{n-1}, \dots, \frac 1hBCH(hV_2, hV_1)\dots)$$
> > While the backward iterates are an approximation of the reversed order.
> >
> >
> > As stated in the introduction, the edge of stability has been established in the full batch regime. In this regime, backward and forward iterations coincide, and our findings are not relevant for this regime.
> >
> > >*Question 9: Clean up typos [...]*
> >
> > Agreed. Thanks.

---

### Review · Reviewer_gbTF · 2025-09-15

**Summary Of Contributions:**

This paper investigates how the composition order of maps affects convergence and stability in optimization. The authors introduce and study a training variant—backward-SGD—in which each iterate is obtained by applying all past batch updates in reverse order rather than the standard forward order. They prove that, under uniform contraction conditions, backward iterations converge to a single point at an exponential rate (Theorem 2.2). Moreover, they establish that the limiting backward points are distributed according to the stationary distribution of the forward process (Theorem 2.7).

Strengths:

1. The idea is conceptually novel. It could have important implications for training stability, early stopping strategies, and hyperparameter tuning.

2. The authors support their claims with clear toy examples, analytical derivations, and extensive empirical validation on real-world applications.

**Audience:**

Yes

**Audience Explanation:**

Yes. The paper’s findings on how iteration order affects convergence and stability would likely interest TMLR readers working on optimization, training dynamics, and algorithmic stability in deep learning.

**Claims And Evidence:**

Yes

**Claims Explanation:**

Yes. The theoretical results are supported by formal proofs and validated through experiments.

**Requested Changes:**

I have the following concerns and questions:

1. Backward SGD is significantly more expensive than forward SGD in computational and memory costs. Even with the approximation proposed in Section C, it still requires computing Hessians.

2. While Theorem 2.2 proves convergence to a point, it does not guarantee that this point is meaningful. In the SGD context, Examples 3.1, 3.2, and 2.4 suggest that the first map $T_1$ has a large influence — the initial error term $h \epsilon_1$ persists and does not decay. By contrast, forward SGD with a diminishing step size can be shown to converge to the true minimizer. For backward SGD, if the first batch is unrepresentative or poor, the algorithm may still converge, but to a suboptimal point. In line with Theorem 2.7, this means that, in terms of performance, backward SGD is not inherently better than forward SGD.

3. Applying Theorem 2.2 to SGD may require restrictive conditions, as noted in Remark 2.3: namely, that each per-batch gradient descent step is a uniform contraction, and that there exists a $\theta^*$ which is a critical point for all $L_i$.

---

> ### Author Response · Authors · 2025-09-26
>
> Thank you for your insightful review. We are very pleased that you found our core idea "conceptually novel" and well-supported by our theoretical and empirical evidence.
> We find ourselves in agreement with your points on practicality and the nature of the convergence point. Your feedback has also motivated us to provide a generalization of our theoretical framing, in appendix G.
>
> Keep in mind that our main goal remains to introduce backward-SGD as a theoretical (and possible diagnostic tool), and not a practical optimizer. We use it to reveal the profound impact of iterate composition order on stability. You’ll find below our best attempt to address your questions.

---

> > ### Author Response · Authors · 2025-09-26
> > **Question 1**
> >
> > >*Backward SGD is significantly more expensive than forward SGD in computational and memory costs. Even with the approximation proposed in Section C, it still requires computing Hessians.*
> >
> > We agree with you. The naive backward-SGD is computationally prohibitive and even the approximation in Appendix C is still too costly for practical use if we use the exact Hessian. However, the use of Hessian approximations (like a diagonal approximation for instance) makes the algorithm in Appendix C more tractable, although we are not pursuing this avenue in this current paper, whose aim is to establish the theoretical foundation of the iteration composition approach. We will revise our paper to make it clearer that backward-SGD is intended as a conceptual tool for analysis. The approximation in Appendix C is presented only as a *first step* and "proof of concept" for a potential *future* research direction aimed at finding cheaper methods to harness the stability properties we identified. We will clarify that developing a truly practical, low-cost algorithm is a significant, open research problem.

---

> > > ### Author Response · Authors · 2025-09-26
> > > **Question 2**
> > >
> > > >*While Theorem 2.2 proves convergence to a point, it does not guarantee that this point is meaningful. In the SGD context, Examples 3.1, 3.2, and 2.4 suggest that the first map  has a large influence — the initial error term  persists and does not decay. By contrast, forward SGD with a diminishing step size can be shown to converge to the true minimizer. For backward SGD, if the first batch is unrepresentative or poor, the algorithm may still converge, but to a suboptimal point. In line with Theorem 2.7, this means that, in terms of performance, backward SGD is not inherently better than forward SGD.*
> > >
> > > This is a good observation, and we agree with you:
> > >
> > > * Backward-SGD is *not inherently better* than forward-SGD in terms of final test performance.
> > >
> > >
> > > * It is highly sensitive to the initial batch, as shown in our Example 2.4. If that batch is "poor," it will converge to a "suboptimal" point.
> > >
> > >
> > > This is, in fact, a key part of our discovery. Our paper aims to show that the composition order creates two different processes, which we could describe intuitively as follows:
> > >
> > > * *Forward-SGD:* Converges to a distribution, "forgetting" the initial batches and being most influenced by recent ones. This makes it robust to a bad start but leaves it perpetually noisy.
> > >
> > > * *Backward-SGD:* Converges to a point, "remembering" the initial batches but "forgetting" the recent stochastic noise.
> > >
> > > As you correctly infer from our Theorem 2.7, any single point sampled from the backward process is drawn from the *same* stationary distribution as the forward process, so there is no "performance" advantage. The *true value* of the backward trajectory is its *stability*. Because it converges to a point, it provides a "clean" or "denoised" view of the optimization path. We will revise our discussion to explicitly frame this as a fundamental trade-off: forward-SGD offers robustness to initialization, while backward-SGD offers trajectory stability.

---

> > > > ### Author Response · Authors · 2025-09-26
> > > > **Question 3**
> > > >
> > > > >*Applying Theorem 2.2 to SGD may require restrictive conditions, as noted in Remark 2.3: namely, that each per-batch gradient descent step is a uniform contraction, and that there exists a  which is a critical point for all.*
> > > >
> > > > These are two good points.
> > > >
> > > > For the latter point, the existence of a global critical point common to each batch loss $L_i$ can be immediately derived from the existence of a global critical point for the full loss $L$, since the $L_i$ can in general be assumed to be non-negative. Now, in the case of large neural networks, overparametrization generally implies the existence of such global minima (actually a high-dimensional submanifold of them) for the full loss $L$.
> > > >
> > > > The former point you are raising is a crucial point about the restrictiveness of the uniform contraction assumption. In the standard Euclidean metric, this is indeed a too strong assumption for non-convex deep learning.
> > > > To address this exact gap, we have
> > > >
> > > > 1. added a new discussion in Appendix G, and
> > > > 1.  updated Remark 2.3 to reflect this new discussion.
> > > >
> > > > In this new section (Appendix G), we show that our backward contraction principle (Theorem 2.2) holds in the more general and relevant setting of deep learning. However, this comes at a much higher technical cost involving the introduction of the abstract notion of pseudo-metric spaces. However, the core of the argument in the proof of Theorem 2.2 remains unchanged, although the convergence is now in abstract “quotient spaces”.
> > > >
> > > >
> > > > Specifically, we show that if the loss satisfies the *Polyak-Lojasiewicz (PL) condition* (which, as argued by Liu et al. (2022), is often satisfied in over-parameterized neural networks) then the gradient updates are *contractions on average* with respect to a "loss pseudo-metric" (defined as $d(x,y)=L(x)+L(y)$). We then explain that our how backward contraction argument (as outlined in the proof of Theorem 2.2) works for this pseudo-metric as well.
> > > >
> > > >
> > > > Therefore, our theoretical results are not limited to a simple convex case but are indeed applicable to the non-convex settings of deep learning, providing a formal explanation for the powerful stabilizing effects we observe in all our experiments.

---

> > > > > ### Comment · Reviewer_gbTF · 2025-10-15
> > > > >
> > > > > Thank you for the response. I understand that the paper is intended as a "proof of concept." I suggest the authors consider incorporating the points made in their response to Question 2 more clearly into the paper.

---

> > > > > > ### Author Response · Authors · 2025-10-16
> > > > > >
> > > > > > Thank you! We now added a paragraph in section 2.2 (in blue) summarizing our answers to question 2, and uploaded a revised version.

---

### Decision · Action_Editor_VmGu · 2025-11-13

**Recommendation:** Accept with minor revision

**Additional Comments:**

I expect the authors to update their paper to fix any remaining issues regarding the concerns from the reviewers, in particular it should be clearly explained in the work that it is meant as a proof of concept, the paper shouldn't be misleading. I also think that the request from Reviewer rt2N to include a similar exploration with AdamW is very reasonable as it's the most commonly used optimizer in practice and it shouldn't be hard to add at least one experiment given that it requires changing just a single line of code to run it.

I also highly encourage the authors to open source their experiments and provide a link to a github repository in their paper. While it is not mandatory, it is important to keep research reproducible, and I hope the authors agree that it's better if their code is available online. It doesn't have to be clean, but it would in the very least serve as a reference for how the experiments were done.

**Audience:**

Yes

**Audience Explanation:**

The topic of training stability is of high relevance to TMLR. While the reviewers questioned the significance of these results, it was clarified in the rebuttal period that the work was meant as a proof of concept. The main idea has also been described by reviewers as "conceptually novel", so I expect it would interest at least some audience of TMLR.

**Claims And Evidence:**

Yes

**Claims Explanation:**

In this work, the authors study the impact of the order of stochastic samples on the stability of SGD. They make the observation that the backward sampling would lead to much more stable trajectories and provide rigorous theoretical proofs as well as back up this numerically. The authors also explicitly state the limitations of their work (computational intensity).